# KScope: A Framework for Characterizing the Knowledge Status of Language Models

**Yuxin Xiao**[1], **Shan Chen**[2], **Jack Gallifant**[2],
**Danielle Bitterman**[2], **Thomas Hartvigsen**[3], **Marzyeh Ghassemi**[1]
[1]Massachusetts Institute of Technology, [2]Harvard University, [3]University of Virginia
`yuxin102@mit.edu`

## Abstract

Characterizing a large language model's (LLM's) knowledge of a given question is challenging. As a result, prior work has primarily examined LLM behavior under knowledge conflicts, where the model's internal parametric memory contradicts information in the external context. However, this does not fully reflect how well the model knows the answer to the question. In this paper, we first introduce a taxonomy of five knowledge statuses based on the consistency and correctness of LLM knowledge modes. We then propose KScope, a hierarchical framework of statistical tests that progressively refines hypotheses about knowledge modes and characterizes LLM knowledge into one of these five statuses. We apply KScope to nine LLMs across four datasets and systematically establish: (1) Supporting context narrows knowledge gaps across models. (2) Context features related to difficulty, relevance, and familiarity drive successful knowledge updates. (3) LLMs exhibit similar feature preferences when partially correct or conflicted, but diverge sharply when consistently wrong. (4) Context summarization constrained by our feature analysis, together with enhanced credibility, further improves update effectiveness and generalizes across LLMs.

## 1   Introduction

LLMs [15, 24, 55, 72] memorize information from their training corpora as *parametric knowledge* [3, 4, 5, 58]. They may further incorporate grounded and up-to-date *contextual knowledge* from user prompts for knowledge-intensive tasks [14, 34, 60]. Knowledge conflicts can arise when an LLM's parametric knowledge contradicts the contextual input [71]. Existing work has sought to measure and understand LLM behavior under these conflicting conditions [11, 30, 39, 68, 69, 76].

However, prior studies on knowledge conflicts do not fully characterize an LLM's underlying knowledge of a given question. Recent work usually represents LLM's knowledge via the most likely response [30, 62, 69], which may overlook the coexistence of multiple competing modes in an answer distribution. Moreover, entropy-based uncertainty metrics [11, 39] capture overall uncertainty instead of mode structure, and would assign similar entropy values (approximately 1.37) to distributions $[0.45, 0.45, 0.1]$ and $[0.6, 0.2, 0.2]$. Yet the first distribution reflects conflicting knowledge, while the second shows consistent preference.

To address this gap, we define a taxonomy of five knowledge statuses along two key dimensions and propose KScope, a hierarchical testing framework for characterizing knowledge status. As shown in Figure 1, we assess *knowledge consistency* by examining the size of an LLM's mode set, and evaluate *knowledge correctness* relative to the ground truth. Based on these two dimensions, we identify five distinct knowledge statuses: (1) consistent correct, (2) conflicting correct, (3) absent, (4) conflicting wrong, and (5) consistent wrong. We construct an empirical response distribution by repeatedly sampling the target LLM, and leverage KScope to progressively refine hypotheses about its underlying knowledge modes.

39th Conference on Neural Information Processing Systems (NeurIPS 2025).

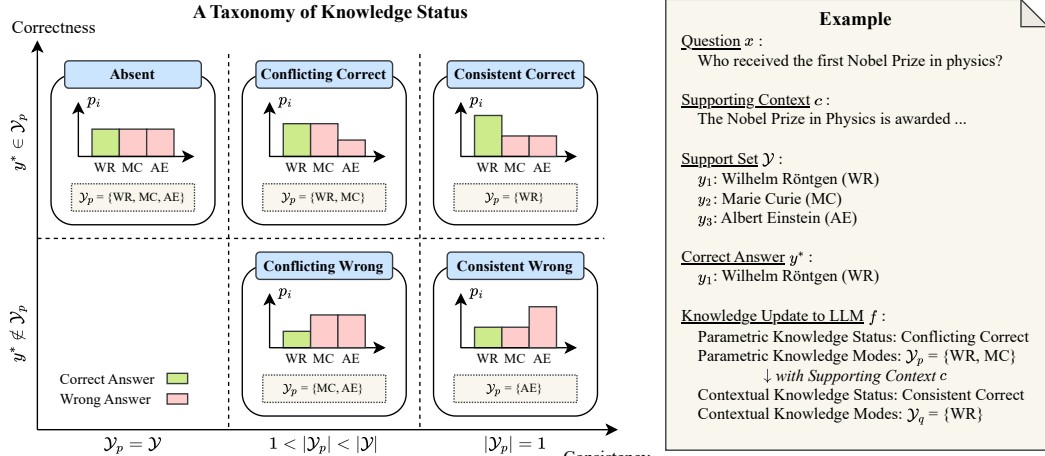

Figure 1: We propose a taxonomy of five knowledge statuses based on the consistency and correctness of LLM knowledge modes. We illustrate the taxonomy using an LLM's parametric knowledge modes $\mathcal{Y}_p$ in a three-option classification task. This formulation also applies to contextual knowledge modes $\mathcal{Y}_q$ and generalizes to open-ended questions or classification tasks with more options.

We evaluate the knowledge status of nine LLMs across four datasets (Section 5), and find that supporting context increases the proportion of consistent correct knowledge across all datasets and models. In the multi-choice setting, we note that the two healthcare-related datasets (Hemonc [64] and PubMedQA [26]) exhibit lower levels of consistent correct knowledge than the general-domain datasets (NQ [32] and HotpotQA [74]), both before and after providing context. Within each model family, larger LLMs exhibit higher proportions of consistent correct parametric knowledge. Among them, Llama-3 [15] achieves the highest proportion, followed by Qwen-2.5 [72] and Gemma-2 [55], although these differences narrow when context is introduced. Finally, we show that noisy retrieval and open-ended question settings substantially reduce the effectiveness of updating LLMs to exhibit consistently correct knowledge.

We next examine what features of the input context successfully update LLM knowledge to the consistent correct status (Section 6). We select eleven context features across three categories—difficulty, relevance, and familiarity, and show that all three feature categories contribute, with context length and entropy consistently valued across statuses. We also find that LLMs prioritize context features similarly when they are partially correct or exhibit conflicting knowledge, regardless of correctness. In contrast, the consistent wrong status shows relatively low correlations with other statuses, suggesting that overcoming strongly held false beliefs may require distinct context features.

Finally, we explore context augmentation strategies to improve the success rate of knowledge updates (Section 7). We find that context summarization with constraints informed by our feature importance analysis outperforms naïve summarization. When combined with enhanced context credibility [70], this approach improves the proportion of successful knowledge updates by $4.3\%$ on average across all statuses—even in GPT-4o [24], which was not included in our feature analysis.

We summarize our contributions[1] in this paper as follows:

- We define a taxonomy of five knowledge statuses based on consistency and correctness, and propose KScope, a hierarchical testing framework to characterize LLM knowledge status.
- We apply KScope to nine LLMs across four datasets, and establish that supporting context substantially narrows knowledge gaps across model sizes and families.
- We identify key context features related to difficulty, relevance, and familiarity that drive successful knowledge updates.
- We reveal how LLM feature importance differs based on parametric knowledge status, showing similarity under conflict but divergence when consistently wrong.
- We validate that constrained context summarization, combined with improved credibility, significantly boosts successful knowledge updates across all statuses and generalizes well.

---

[1]Our code is available at `https://github.com/xiaoyuxin1002/KScope`.

| Step | Statistical Test | Null Hypothesis | Alternative Hypothesis | If Significant $p$-value | If Insignificant $p$-value |
|---|---|---|---|---|---|
| (1) Test for the Significance of Invalid Answers | One-Sided Exact Binomial Test | $\mathbb{P}(f(x) \in \mathcal{Y}) = \mathbb{P}(f(x) \notin \mathcal{Y}) = \frac{1}{2}$ | $\mathbb{P}(f(x) \notin \mathcal{Y}) > \frac{1}{2}$ | Absent Knowledge | Proceed ↓ |
| (2) Test for Uniform Guessing | Two-Sided Exact Multinomial Test | $p_i = \frac{1}{|\mathcal{Y}|}, \forall y_i \in \mathcal{Y}$ | $p_i \neq \frac{1}{|\mathcal{Y}|}, \exists y_i \in \mathcal{Y}$ | Proceed ↓ | Absent Knowledge |
| (3) Test for Conflicting Knowledge | Likelihood Ratio Test | $p_i = \frac{1}{|\mathcal{Y}|}, \forall y_i \in \mathcal{Y}$ | (a) $p_1 = p_2 = \frac{\hat{p}_1 + \hat{p}_2}{2} > p_3 = \hat{p}_3$ 
 (b) $p_1 = p_3 = \frac{\hat{p}_1 + \hat{p}_3}{2} > p_2 = \hat{p}_2$ 
 (c) $p_2 = p_3 = \frac{\hat{p}_2 + \hat{p}_3}{2} > p_1 = \hat{p}_1$ | Proceed Accordingly ↓ | Absent Knowledge |
| (4) Test for Consistent Knowledge | One-Sided Exact Binomial Test | $p_1' = p_2' = \frac{1}{2}$ | (a) $p_1' = \frac{\hat{p}_1}{\hat{p}_1 + \hat{p}_2} > p_2' = \frac{\hat{p}_2}{\hat{p}_1 + \hat{p}_2}$ 
 (b) $p_1' = \frac{\hat{p}_1}{\hat{p}_1 + \hat{p}_2} < p_2' = \frac{\hat{p}_2}{\hat{p}_1 + \hat{p}_2}$ | Consistent Correct / Wrong Knowledge (depending on correctness) | Conflicting Correct / Wrong Knowledge (depending on correctness) |

Figure 2: We propose KScope, a hierarchical testing framework to characterize LLM knowledge into one of the five identified statuses. We note that our framework generalizes to questions with larger support sets by repeating Step 3 to iteratively refine hypotheses about the knowledge mode set.

## 2 Related Work

**Knowledge Characterization.** LLMs have been shown to memorize world knowledge from their training corpora [3, 4, 5, 10, 44, 47, 58]. Existing work interprets this memorization mechanism by probing activation patterns [7, 8, 23, 33] and benchmarks the factual accuracy of recalled information against knowledge graphs [36, 78, 82]. Researchers have employed multi-round prompting to improve the consistency [43, 61, 77] and calibration [19, 31, 73] of the knowledge elicited from LLMs. In this work, we identify five knowledge statuses that an LLM may exhibit with respect to a given question and explicitly characterize them through a hierarchical statistical testing framework.

**Knowledge Conflict.** Knowledge conflict can arise within an LLM's parametric memory, within the provided context, or between the two [71]. Prior work has examined various contextual factors that influence a model's degree of compliance [53, 54, 57, 70] and introduced metrics to quantify the persuasiveness of context [11, 39]. These studies find that LLMs tend to follow the context when they are uncertain [45, 68, 76] or when there is confirmation bias between the model's memory and the context [30, 69]. Large-scale evaluation benchmarks and pipelines have also been developed to facilitate research in this area [20, 51, 62]. However, prior work usually uses a single sampled response to represent model memory [30, 62, 69] and applies entropy-based measures [11, 39] to assess conflict, both of which overlook the mode structure of the response distribution. In contrast, we define five knowledge statuses to measure the consistency and correctness of knowledge, rigorously test different mode structures, and stratify our analysis of model behavior based on these statuses.

**Knowledge Update.** To incorporate accurate and up-to-date information, retrieval-augmented generation (RAG) systems [6, 14, 17, 34, 46, 49] retrieve relevant context from external corpora to support LLMs in knowledge-intensive tasks. The retrieved context can be further augmented to enhance the effectiveness of knowledge updates [53, 57, 70]. Other approaches directly edit a model's parametric knowledge [12, 18, 59, 60] or apply mechanistic interventions at inference time [27, 35, 80]. In this work, we examine knowledge updates under both gold and noisy retrieval settings and systematically evaluate various context augmentation strategies tailored to each knowledge status.

## 3 LLM Knowledge Status and How to Characterize it

### 3.1 LLM Knowledge Status: Consistency and Correctness

When analyzing an LLM's knowledge, we focus on two key dimensions:

1. **Consistency**: How consistent are the model's knowledge modes? That is, does it exhibit a single coherent belief or multiple conflicting ones?
2. **Correctness**: Does the set of the model's knowledge modes include the correct answer?

To formalize this, consider a question-answer pair $(x, y^*)$. We define the parametric knowledge of an LLM $f$, with respect to the question $x$, as the conditional multinomial distribution $\mathbf{p} = f(\cdot \mid x)$ over the support set $\mathcal{Y} = \{y_1, \ldots, y_d\}$, where $p_i = f(y_i \mid x)$. We further define the set of parametric

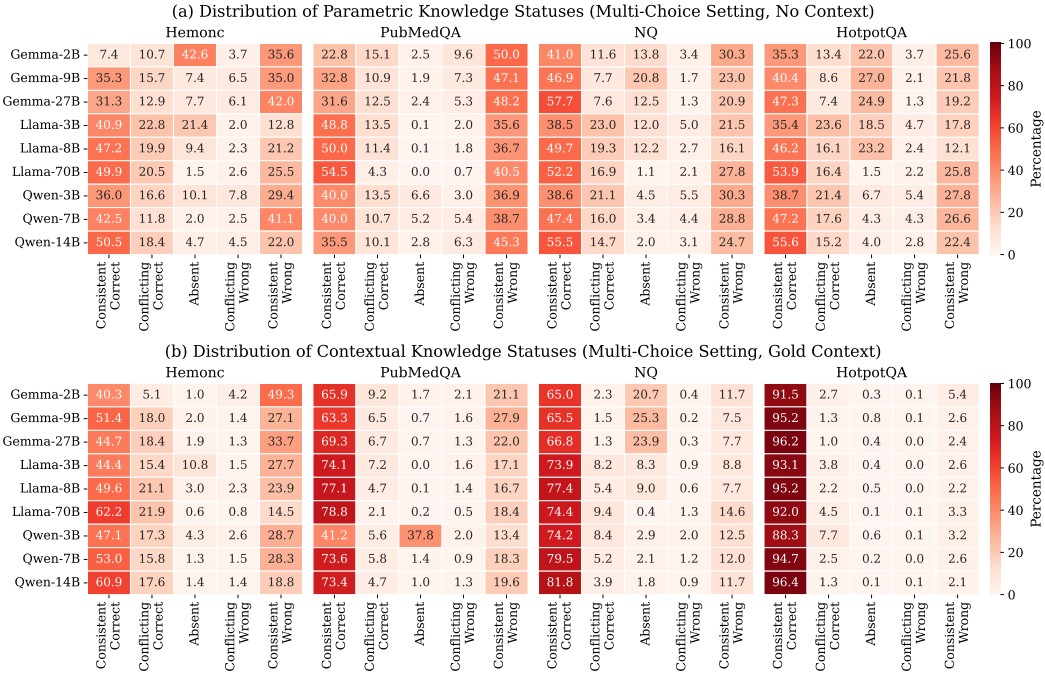

Figure 3: Characterization results from applying the KScope framework to nine LLMs across four datasets. Overall, most LLMs exhibit the highest proportion of consistent correct parametric knowledge status, which further increases when gold context is provided.

knowledge modes $\mathcal{Y}_p = \text{modes}(\mathbf{p}) \subseteq \mathcal{Y}$ as the subset that satisfies the following condition: $p_i = p_j$ for any $y_i, y_j \in \mathcal{Y}_p$ and $p_i > p_k$ for any $y_k \notin \mathcal{Y}_p$. In essence, the knowledge modes form a plateau of high-probability elements within the support set that are distinguishable from the rest.

To assess the consistency dimension, we examine three possible structures of $\mathcal{Y}_p$: (1) $\mathcal{Y}_p = \mathcal{Y}$, (2) $1 < |\mathcal{Y}_p| < |\mathcal{Y}|$, and (3) $|\mathcal{Y}_p| = 1$, where $|\cdot|$ denotes the cardinality of a set. We evaluate correctness by checking whether the ground-truth answer $y^* \in \mathcal{Y}_p$. Based on these two dimensions, we formulate a taxonomy of five parametric knowledge statuses $P = \text{status}(\mathcal{Y}_p)$, as illustrated in Figure 1, that characterize the knowledge of $f$ with respect to the question $x$:

1. **Consistent Correct Knowledge**: $|\mathcal{Y}_p| = 1$ and $y^* \in \mathcal{Y}_p$.
2. **Conflicting Correct Knowledge**: $1 < |\mathcal{Y}_p| < |\mathcal{Y}|$ and $y^* \in \mathcal{Y}_p$.
3. **Absent Knowledge**: $\mathcal{Y}_p = \mathcal{Y}$.
4. **Conflicting Wrong Knowledge**: $1 < |\mathcal{Y}_p| < |\mathcal{Y}|$ and $y^* \notin \mathcal{Y}_p$.
5. **Consistent Wrong Knowledge**: $|\mathcal{Y}_p| = 1$ and $y^* \notin \mathcal{Y}_p$.

When supporting context $c$ is available for the question-answer pair, we define the LLM's contextual knowledge as $\mathbf{q} = f(\cdot \mid x, c)$. Analogously, we assign its contextual knowledge status $Q = \text{status}(\mathcal{Y}_q)$ based on the corresponding knowledge modes $\mathcal{Y}_q = \text{modes}(\mathbf{q}) \subseteq \mathcal{Y}$.

### 3.2 Challenges in Operationalizing Knowledge Status

While the taxonomy introduced in Section 3.1 offers a principled view of LLM knowledge status, applying it in practice poses several challenges. First, the true underlying distributions of LLM knowledge are unobservable. Second, even under the same knowledge status, models may behave differently. For instance, when an LLM lacks sufficient knowledge about a question, it may either respond randomly or refuse to answer altogether [22, 66].

To address the first challenge, we approximate the latent knowledge distributions using empirical sample frequencies. Specifically, we first generate $M$ paraphrases of a given question to reduce prompt sensitivity [48], then collect $N$ chain-of-thought responses [65] from the target LLM using these paraphrases. For open-ended generation, we define the support set $\mathcal{Y}$ by semantically clustering the $N$ samples [31]. For multiple-choice tasks, $\mathcal{Y}$ is simply the set of given options.

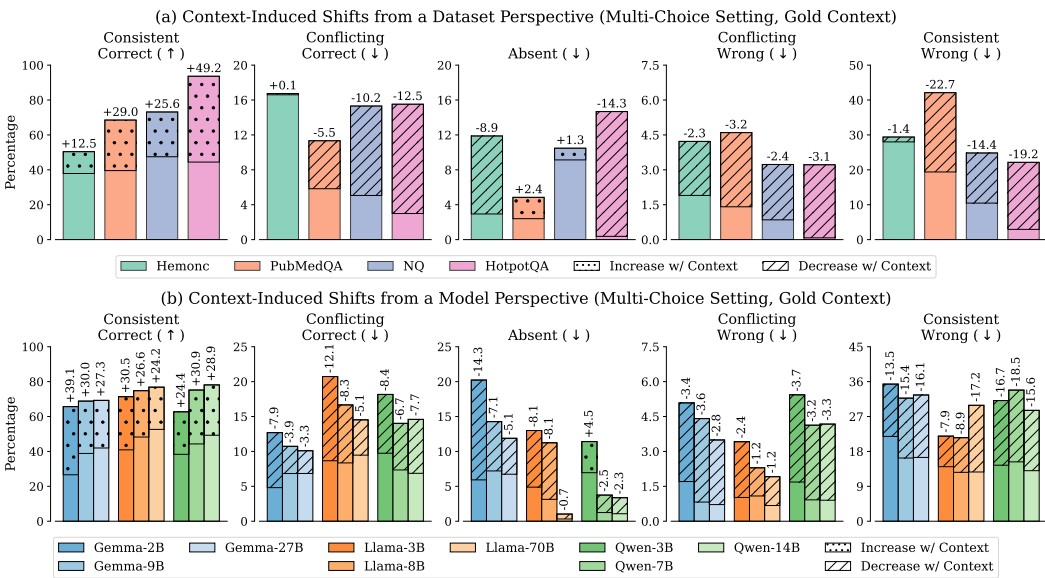

Figure 4: Context-induced shifts in knowledge status distributions. Supporting context increases the proportion of consistent correct knowledge across all datasets and models. The Llama family and larger models within each family achieve higher proportions of consistent correct knowledge, although the gaps narrow with context.

We then account for the second challenge by considering the possibility of invalid model responses. These include hallucinating answers outside the support set [22] or refusing to respond in high-stakes applications [66]. Let $N'$ denote the number of invalid responses. We estimate the empirical distribution $\hat{\mathbf{p}}$ (and analogously, $\hat{\mathbf{q}}$) by: $\hat{p}_i = \frac{1}{N-N'} \sum_{n=1}^{N} \mathbb{1}[f_n(x) = y_i], \forall y_i \in \mathcal{Y}$.

### 3.3 KScope: Knowledge Status Characterization via Hierarchical Testing

To bridge the gap between the true latent knowledge distributions discussed in Section 3.1 and the empirical distributions estimated in Section 3.2, we introduce **KScope**, a hierarchical testing framework for knowledge status characterization. We illustrate the testing details in Figure 2.

**Step 1: Test for the Significance of Invalid Answers.** We first assess whether the model exhibits a higher tendency to produce invalid responses via a one-sided exact binomial test.

**Step 2: Test for Uniform Guessing.** Next, we test whether the LLM's empirical response distribution significantly deviates from a uniform distribution using a two-sided exact multinomial test.

**Step 3: Test for Conflicting Knowledge.** Subsequently, we perform a set of likelihood ratio tests to refine the model's knowledge mode set. Alternatives whose estimated probabilities violate their own inequality constraints are immediately rejected. If multiple alternatives remain significant after Bonferroni correction, we select the one with the lowest Bayesian Information Criterion (BIC). For larger support sets, we repeat this step to remove low-probability elements from the mode set.

**Step 4: Test for Consistent Knowledge.** The previous step reduces the model's knowledge mode set to two elements. Conditioned on the selected alternative in Step 3, we then test whether the model assigns significantly different probabilities to the two remaining elements using two one-sided exact binomial tests. As before, we discard invalid alternatives and, if multiple alternatives are accepted, select the one with the lowest BIC.

## 4 Experiment Setup

**Datasets.** We focus on four tasks, two from the healthcare domain and two from the general domain:

- **Hemonc** [64] is a healthcare dataset extracted from a regularly maintained oncology reference database. It consists of 6,212 clinical study instances, each comparing the efficacy of a regimen versus a comparator for a given medical condition, labeled as superior, inferior, or

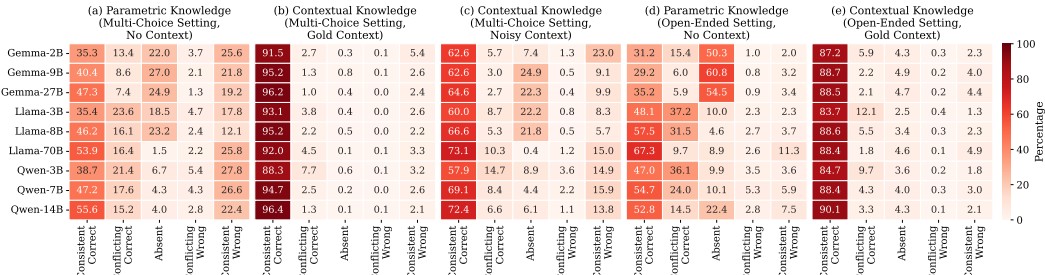

Figure 5: Characterization results from applying KScope to nine LLMs on HotpotQA across different settings. Compared to (b), where gold context in the multi-choice setting enables more consistent correct knowledge, (c) noisy context and (e) the open-ended setting yield lower update success. Without context, the Gemma family shows more absent knowledge in (d) the open-ended setting than in (a) the multi-choice setting, whereas the Llama and Qwen families mostly show the opposite trend.

no difference. To reduce positional bias, we permute the order of each regimen–comparator pair. The context for each question consists of abstracts from the associated PubMed articles.

- **PubMedQA** [26] consists of 1,000 medical research questions, each labeled with a yes, no, or maybe answer. The context comes from the corresponding PubMed abstracts.
- **NQ** [32] contains 3,596 Google search queries, retrieving Wikipedia pages as context.
- **HotpotQA** [74] contains 6,119 multi-hop reasoning questions in the general domain, using sentence-level supporting facts extracted from relevant Wikipedia articles as context.

Unless otherwise specified, all the supplied contexts here contain ground-truth evidence. When comparing LLMs' knowledge statuses in the multi-choice setting, we convert NQ and HotpotQA into three-option classification tasks. Following [70], we prompt GPT-4o [24] to generate two additional wrong options for each question (details in Appendix A). We note that although not evaluated here, KScope remains applicable to classification with more options, as discussed in Section 3.

**Implementation Details.** We evaluate nine instruction-tuned LLMs spanning three model families: Gemma-2 (2B, 9B, 27B) [55], Llama-3 (3B, 8B, 70B) [15], and Qwen-2.5 (3B, 7B, 14B) [72]. We keep the LLMs' sampling distributions unchanged by setting the temperature to 1, and fix the significance level used in KScope at $\alpha = 0.05$. A hyperparameter search on Hemonc using Llama-8B (details in Appendix A) shows that knowledge status characterization empirically stabilizes after $N = 100$ samples, using $M = 20$ paraphrases per question.

## 5 Q1: How Does Context Update LLMs' Knowledge Status?

Using the setup in Section 4, we first characterize the distributions of LLMs' parametric and contextual knowledge statuses in the multi-choice setting and examine how gold context induces shifts between them in Section 5.1. We further investigate the effects of noisy context in Section 5.2 and analyze knowledge status distributions in the open-ended setting in Section 5.3.

### 5.1 Knowledge Status in the Multi-Choice Setting with Gold Context

We apply KScope to the nine LLMs across the four datasets in the multi-choice setting and present the results in Figure 3. When relying solely on parametric knowledge (Figure 3 (a)), LLMs exhibit consistent knowledge—whether correct or wrong—more frequently than conflicting knowledge. When conflicting knowledge occurs, the correct answer is usually among the knowledge modes. Some outliers show a higher proportion of absent knowledge, such as Gemma-2B on Hemonc.

When LLMs are provided with gold context (Figure 3 (b)), the proportion of consistent correct knowledge status significantly increases across all datasets and models, with the largest improvement in HotpotQA and the smallest in Hemonc. This highlights the effectiveness of retrieval-augmented generation [6, 14, 17, 34], which aims to enhance LLMs' knowledge with relevant external information. However, in some cases, context may confuse LLMs, leading them to guess randomly and increasing the proportion of absent knowledge. For example, this occurs with Qwen-3B on PubMedQA and the Gemma family on NQ, likely due to longer context lengths in these datasets—an issue we further investigate in Section 6.

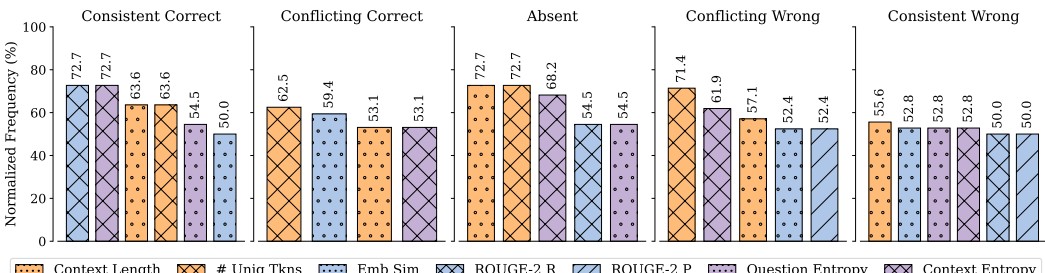

Figure 6: Context features appearing among the top five most important in at least $50\%$ of the cases for each parametric knowledge status. Each color indicates a feature category: orange for difficulty, blue for relevance, and purple for familiarity. Across all statuses, context length and entropy consistently rank among the most important features.

We further examine how gold context shifts the distribution of each knowledge status. Figure 4 (a) presents dataset-level shifts, where distributions of knowledge status are averaged across the nine LLMs. We observe that Hemonc and PubMedQA exhibit lower proportions of consistent correct knowledge than NQ and HotpotQA, both before and after context is provided. Additionally, the gold context slightly increases the ratio of conflicting correct knowledge in Hemonc and absent knowledge in PubMedQA, highlighting the challenges introduced by medical-domain context [9, 50, 56, 63].

Within each LLM family, we find that larger models consistently exhibit higher proportions of consistent correct knowledge, both before and after providing context (Figure 4 (b)) [5, 37]. Llama achieves the highest proportion of consistent correct parametric knowledge, followed by Qwen and Gemma, although this gap narrows once context is introduced. Appendix B details how gold context shifts each LLM's knowledge among different statuses on each dataset in the multi-choice setting.

## 5.2 Knowledge Status in the Multi-Choice Setting with Noisy Context

To investigate LLMs' knowledge status under more realistic retrieval conditions, we apply KScope to the nine LLMs using the top ten Wikipedia paragraphs retrieved for each HotpotQA question, which may or may not include the gold supporting context. As shown in Figure 5 (c), noisy context results in a much lower success rate of updating models to consistent correct knowledge compared to gold context in (b), highlighting the importance of retrieval quality in RAG systems [6, 14, 17, 34, 46, 49]. When the retrieved noisy context lacks evidence for the ground-truth answer, models either refuse to answer, leading to more absent knowledge, or are misled into producing consistently incorrect answers. More details on noisy context-induced shifts in knowledge status are in Appendix B.

## 5.3 Knowledge Status in the Open-Ended Setting with Gold Context

To demonstrate the generalizability of KScope to open-ended questions [28, 67], we apply it to characterize the knowledge status of the nine LLMs on HotpotQA, this time without providing any pre-defined answer options. Specifically, we generate responses per question and semantically cluster [31] them using gemma-2-9b-it [55]. Based on the size of clusters and the number of invalid answers, we follow the procedure in Section 3.3 to infer the LLM's knowledge status.

As shown in Figure 5 (d), knowledge status distributions vary notably across model families. Without pre-defined options or contextual support, the Gemma family often refuses to answer, leading to a higher proportion of absent knowledge compared to the multi-choice setting in (a). In contrast, Llama and Qwen show a substantial increase in the consistent correct knowledge under these conditions, with the exception of Qwen-14B. Gold context still significantly boosts consistent correct knowledge in the open-ended setting in (e), though the improvement is smaller than in the multi-choice setting in (b). Appendix B provides more details on the shift in knowledge status in the open-ended setting.

## 6 Q2: What Context Features Drive the Desired Knowledge Update?

Based on the results in Section 5, we seek to understand what context features drive increases in consistent correct knowledge. We introduce three categories of context features in Section 6.1, and inspect feature importance for each knowledge status in Section 6.2. To enable direct comparison across datasets, we focus on the multi-choice setting with gold context in the following sections.

## 6.1 Context Features

We consider eleven context features across three categories:

- **Difficulty**: (1) **Context Length**; (2) **Readability**, measured by the Flesch-Kincaid readability test [29], which is based on the average number of words per sentence and syllables per word; and (3) **Number of Unique Tokens** (# Uniq Tkns) after lemmatization.
- **Relevance**: (1) **Embedding Similarity** (Emb Sim), computed as the cosine similarity between the embeddings of each question and its corresponding context using text-embedding-3-large [40]; and (2–4) **ROUGE-2 Recall**, **Precision**, and **F1** (ROUGE-2 R/P/F), based on the bigram overlap between each question and its corresponding context.
- **Familiarity**: (1–2) **Question** and **Context Perplexity**, and (3–4) **Question** and **Context Entropy**, as measured by each LLM.

We define a binary label indicating whether context successfully updates an LLM's parametric knowledge to the consistent correct status. As described in Section 4, for each combination of dataset, LLM, and initial parametric knowledge status, we formulate a stratified binary classification task. We apply logistic regression with L2 regularization to the extracted features and discard cases where performance does not exceed a dummy baseline in Macro-F1, due to extreme class imbalance. Implementation details and full regression results are in Appendix C.

We compute feature importance by averaging the absolute SHAP values [38] of each feature within each stratified combination. We then calculate the normalized frequency with which each feature appears among the top five most important features across datasets and LLMs. This yields a frequency-based ranking for each initial parametric knowledge status.

## 6.2 Feature Importance for Context-Driven Knowledge Update

We plot in Figure 6 the context features that appear among the top five most important in at least 50% of the cases identified in our experiments. The results include features from all three categories: difficulty, relevance, and familiarity. This aligns with [11, 57], which find that context relevance influences context persuasiveness. Among the five parametric knowledge statuses, the consistent wrong knowledge status exhibits a flatter distribution of top feature frequencies compared to the others, suggesting that LLMs in this status place less emphasis on any specific feature. Notably, context length and entropy consistently rank high across all statuses.

To assess whether LLMs in distinct parametric knowledge statuses prioritize context features similarly, we compute Spearman rank correlations with Bonferroni correction from feature importance rankings. As shown in Figure 7, correlations are statistically significant between consistent correct and both conflicting correct and absent knowledge. This suggests a confirmation bias [30, 69]: when context at least partially aligns with the model's knowledge modes, the model attends to context features similarly. We also observe a significant correlation between conflicting correct and conflicting wrong, indicating similar feature preferences during knowledge conflict [45, 68, 76], regardless of correctness. In contrast, the consistent wrong status shows relatively low correlations with others, implying that overcoming a firmly held wrong belief may require different context features.

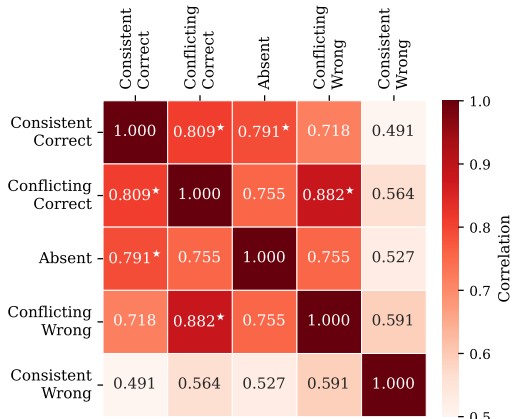

Figure 7: Spearman rank correlation among parametric knowledge statuses based on feature importance rankings. An asterisk indicates statistical significance at a Bonferroni-adjusted $\alpha_{\text{adj}} = 0.005$.

## 7 Q3: What Context Augmentations Work Best Across Knowledge Statuses?

In this section, we leverage the insights from our feature importance analysis in Section 6 to improve knowledge updates in LLMs. We again focus on the multi-choice setting with gold context.

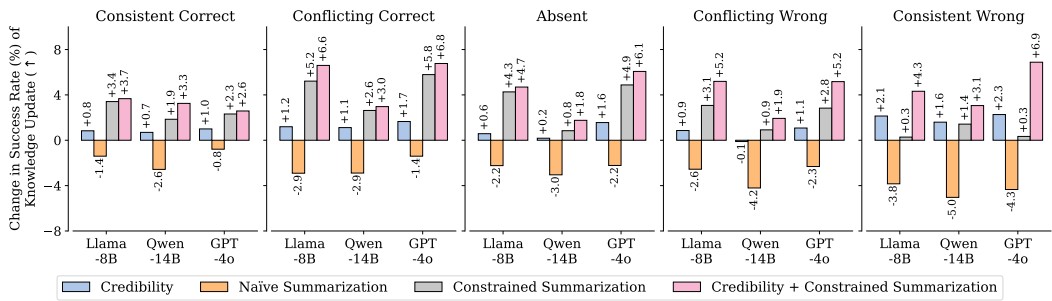

Figure 8: Absolute change (%) in the success rate of knowledge updates for each context augmentation strategy, relative to the original context and averaged across datasets. Integrating credibility metadata into constrained summarization improves the success rate by 4.3% on average across LLMs and statuses, and generalizes well to GPT-4o.

## 7.1 Context Augmentation Strategies

We apply the following augmentation strategies to the original context of the datasets in Section 4:

- **Credibility** [70]: To enhance context credibility, we append the relevant PubMed metadata to the context for Hemonc and PubMedQA, and the corresponding Wikipedia article titles for NQ and HotpotQA. When sampling responses with context, we instruct LLMs to prioritize the credible context over their internal parametric knowledge (details in Appendix D).
- **Naïve Summarization**: We leverage GPT-4o [24] to directly summarize context.
- **Constrained Summarization**: We guide summarization with additional constraints to adjust the context features according to our analysis results. Specifically, we prompt GPT-4o to reduce both context length and the number of unique tokens during summarization. Additionally, GPT-4o is instructed to preserve the semantic content (maintaining embedding similarity with the question), retain token-level overlap (maintaining ROUGE-2 recall and precision), and ensure fluency (preserving context perplexity and entropy).
- **Combined**: We integrate the credibility information into the context generated from constrained summarization.

We then investigate how each augmentation strategy affects the success rate of knowledge updates compared to the original context, for each knowledge status. We evaluate these strategies on three LLMs of varying sizes and families: Llama-3.1-8B-Instruct [15] (Llama-8B), Qwen2.5-14B-Instruct [72] (Qwen-14B), and GPT-4o. Although GPT-4o was not included in the feature analysis in Section 6, we assess it here to examine whether our findings generalize to other LLMs. We present the average results across the four datasets in Figure 8 and the full details in Appendix D.

## 7.2 Effectiveness of Context Augmentations

As shown in Figure 8, constrained summarization improves the success rate across all knowledge statuses except the consistent wrong status. This aligns with our finding in Figure 7 that correcting a consistent wrong belief may require different context features than in other cases. Furthermore, this constrained summarization strategy generalizes well to GPT-4o.

In contrast, naïve summarization always hurts the performance. We plot in Figure 9 the normalized feature space for the nine affected features on Hemonc (question perplexity and entropy remain unchanged by context summarization). Both summarization methods reduce context length and the number of unique tokens, while increasing context perplexity and entropy. However, naïve summarization fails to preserve fluency and key semantic content, resulting in harder readability and lower ROUGE-2 recall. In comparison, constrained summarization improves embedding similarity, ROUGE-2 precision, and F1 more effectively. These differences in the augmented feature space explain the gap in success rates and underscore the importance of our feature analysis in Section 6. Details for other datasets are in Appendix D.

On the other hand, credibility is more effective for the consistent wrong status, as illustrated in Figure 8. This suggests that when an LLM consistently holds a wrong belief, adding credibility metadata to context makes it more persuasive [70]. The combined strategy retains the benefits

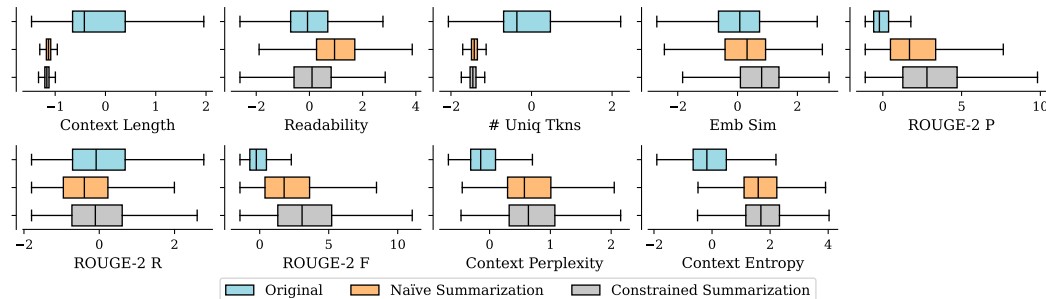

Figure 9: Normalized feature space for original and summarized contexts of Hemonc, measured by Llama-3.1-8B-Instruct. Naïve summarization hurts readability and ROUGE-2 recall, while constrained summarization yields higher embedding similarity, ROUGE-2 precision, and F1.

of constrained summarization for the first four statuses and further improves the success rate for consistent wrong, beyond what credibility alone achieves. Overall, it enhances the success rate by $4.3\%$ on average across all statuses and LLMs, compared to using the original context.

# 8  Discussion

**Conclusion.** In this paper, we first propose a taxonomy of five knowledge statuses based on consistency and correctness. We then introduce KScope, a hierarchical testing framework that characterizes knowledge status by progressively refining hypotheses about an LLM's knowledge modes. By applying KScope to nine LLMs across four datasets, we establish: (1) Supporting context substantially narrows knowledge gaps across LLMs. (2) Features related to context difficulty, relevance, and familiarity drive successful knowledge updates. (3) LLMs attend to features similarly when their knowledge modes are partially aligned with the correct answer or internally conflicting, but diverge sharply when consistently wrong. (4) Constrained context summarization guided by feature analysis, combined with enhanced credibility, further boosts update effectiveness and generalizes across models. These findings provide valuable insights into LLMs' knowledge mechanism [58] and underscore the importance of tailoring knowledge update strategies [14, 60] to different knowledge statuses in future work.

**Limitations.** Our feature importance analysis in Section 6 focuses on eleven features across three categories. However, it remains underexplored how more nuanced stylistic context features [11, 57, 70] impact LLM knowledge status. Although the KScope framework supports classification tasks with any number of options [52, 81], we restrict our experiments to three-option classification due to computational constraints. We also experiment with real-world noisy retrieval, but such context can include conflicting information [20, 51] in practical RAG systems [6, 14, 17, 34, 46, 49], posing additional challenges. We leave the investigation of how context affects LLM knowledge status under these more complex conditions to future work.

**Broader Impacts.** LLMs are widely deployed in everyday user–chatbot applications [1, 24] and high-stakes domains such as healthcare [42, 56] and legal services [13, 16]. However, prior work [11, 39, 53, 54, 57, 70, 71] lacks a formal framework for characterizing LLM knowledge status, especially as these statuses may shift in response to varied input context. The challenge becomes more concerning when training data contains misinformation [2, 41], leading models to develop consistently wrong beliefs. Our analysis in Section 6 shows that correcting these beliefs often requires different context features than those needed for other knowledge statuses. The proposed KScope framework also relates to hallucination detection [22, 25, 79] and uncertainty quantification [21, 31, 75] in LLMs. By identifying knowledge status, it helps distinguish between hallucinations due to absent knowledge and uncertainty due to knowledge conflicts. These connections underscore the practical utility of KScope and its broader impact on improving LLM reliability.

## Acknowledgements

YX is supported by the MIT IDSS Fellowship. MG is supported in part by a National Science Foundation (NSF) 22-586 Faculty Early Career Development Award (#2339381), a Gordon & Betty Moore Foundation award, a Google Research Scholar award, and the AI2050 Program at Schmidt Sciences. SC, JG, and DB are supported by the National Institutes of Health National Cancer Institute (U54CA274516-01A1 [SC, D.S.B.], R01CA294033-01 [JG, D.S.B.]), the American Cancer Society and American Society for Radiation Oncology ASTRO-CSDG-24-1244514-01-CTPS Grant (DOI: `https://doi.org/10.53354/ACS.ASTRO-CSDG-24-1244514-01-CTPS.pc.gr.222210`) [D.S.B.], a Patient-Centered Outcomes Research Institute (PCORI) Project Program Award (ME-2024C2-37484) [D.S.B.], and the Woods Foundation [D.S.B.]. All statements in this report, including its findings and conclusions, are solely those of the authors and do not necessarily represent the views of the sponsors, and no official endorsement should be inferred.

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

# A  Additional Details on the Experiment Setup

In Section 4, we describe the experiment setup, where we apply KScope to nine LLMs across four datasets. Figure 20 shows the few-shot prompt used with GPT-4o [24] to generate two additional incorrect options of the same type as the correct one for each question in NQ [32] and HotpotQA [74]. Figure 21 shows the instructions used to sample model responses with and without context, which are then used by KScope to characterize knowledge statuses.

Among the three existing datasets, PubMedQA [26] uses the MIT license, while NQ and HotpotQA are released under the Apache 2.0 license. All three evaluated LLM families (Gemma [55], Llama [15], and Qwen [72]) are distributed under custom commercial licenses. We run all the experiments in this paper on four NVIDIA A100 GPUs.

To determine the number of question paraphrases ($M$) and sample responses ($N$) needed for consistent characterization of LLM knowledge status, we conduct a hyperparameter search on Hemonc [64] using Llama-8B. As shown in Figure 10, the percentage of status changes stabilizes after collecting $N = 100$ model responses using $M = 20$ paraphrases per question. We adopt this configuration for KScope throughout the paper.

# B  Additional Results on the Context-Induced Shifts in Knowledge Status

In Section 5, we investigate how context updates LLMs' knowledge status. Figures 14, 15, 16, and 17 provide detailed breakdowns of how gold context induces shifts from each parametric knowledge status to contextual knowledge status for each LLM on each dataset in the multi-choice setting. Figure 18 shows the shifts induced by noisy context on HotpotQA in the multi-choice setting, while Figure 19 illustrates the shifts induced by gold context on HotpotQA in the open-ended setting.

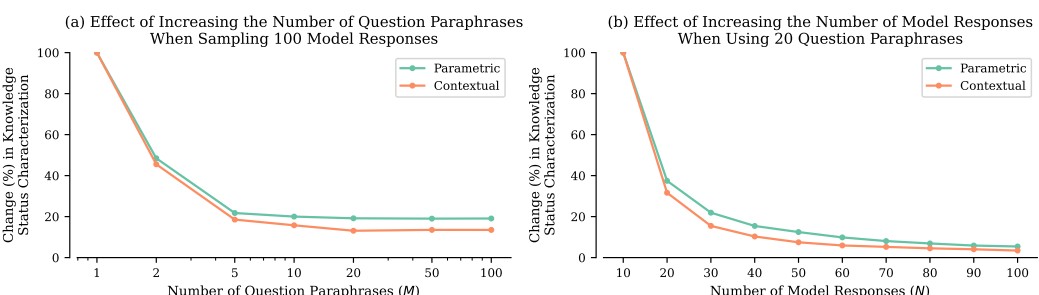

Figure 10: Hyperparameter search on Hemonc using Llama-8B with gold context in the multi-choice setting. KScope achieves stable characterization of LLM knowledge status with $M = 20$ question paraphrases (left) and $N = 100$ model responses per question (right). We adopt this configuration in all experiments.

# C   Additional Results of the Feature Importance Analysis

To identify the context features driving successful knowledge updates, we apply logistic regression to the features extracted in Section 6. We use L2 regularization to mitigate multicollinearity and normalize all numerical features before model fitting. For each stratified combination of dataset, LLM, and initial parametric knowledge status, we perform five-fold cross-validation to tune class weights and regularization strength. We exclude combinations with fewer than 50 examples or 10 instances per class. Due to extreme class imbalance in some settings (e.g., nearly 99% positive labels for Gemma-9B with a consistent correct status on HotpotQA), logistic regression does not always outperform a dummy classifier in Macro-F1, which simply predicts the majority class. We retain only regression models that outperform this baseline for the feature importance analysis in Section 6. Figure 11 shows the change in Macro-F1 relative to the dummy classifier.

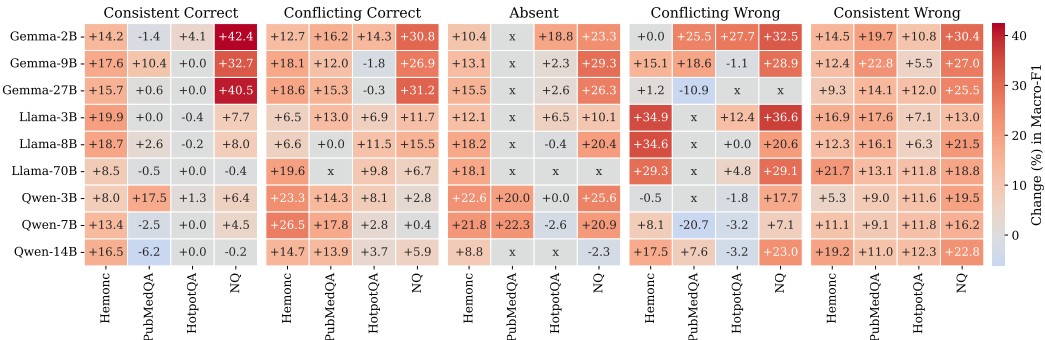

Figure 11: Change in Macro-F1 of logistic regression models relative to a dummy classifier that predicts the majority class. "x" marks cases with fewer than 50 examples or 10 instances per class, which are excluded from regression analysis.

# D  Additional Results on the Context Augmentation Strategies

In Section 7, we experiment with various context augmentation strategies. Figure 22 illustrates how we insert metadata to enhance context credibility. Figure 23 shows how we prompt GPT-4o to perform naïve and constrained context summarization.

We present the normalized feature space for each dataset in Figure 12, highlighting the differences between naïve and constrained summarization. We also show how each augmentation strategy impacts the success rate of knowledge updates for each dataset in Figure 13.

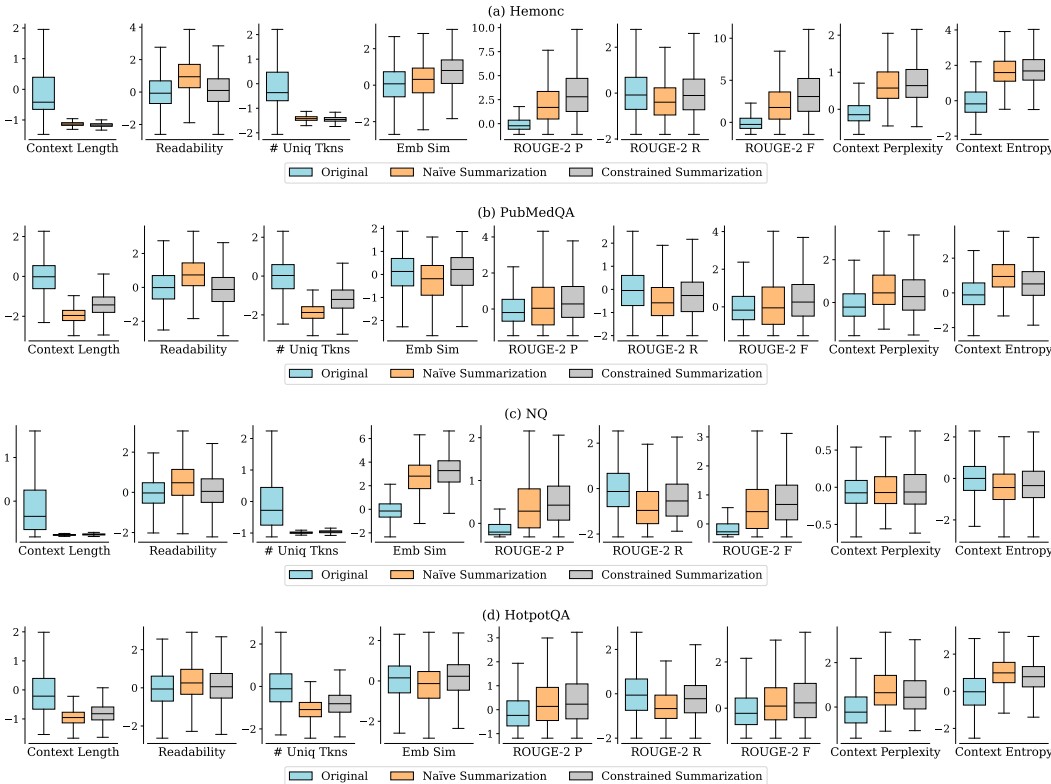

Figure 12: Normalized feature space per dataset, comparing naïve and constrained summarization.

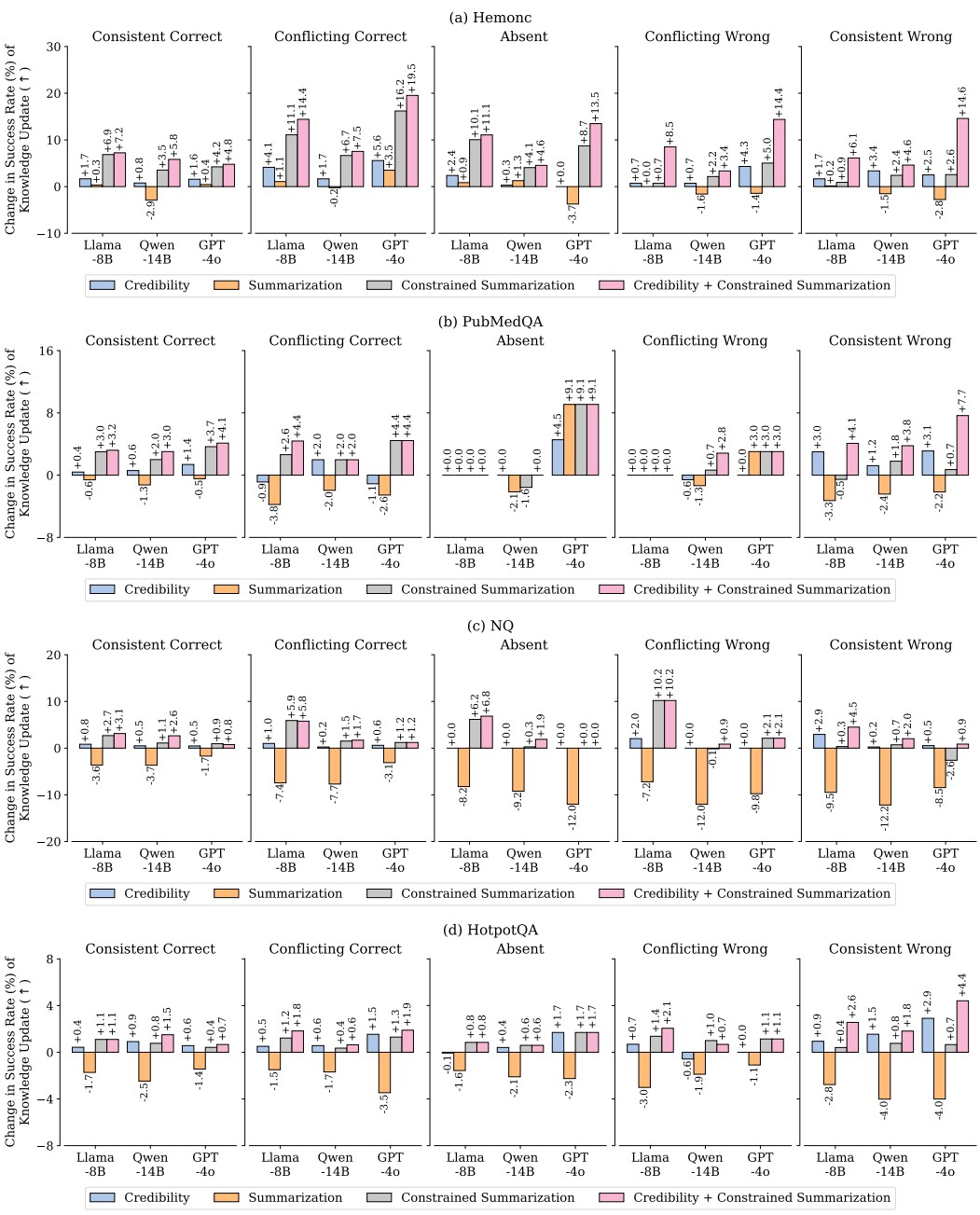

Figure 13: Effect of different context augmentation strategies on the success rate of knowledge updates for each dataset, relative to the original context.

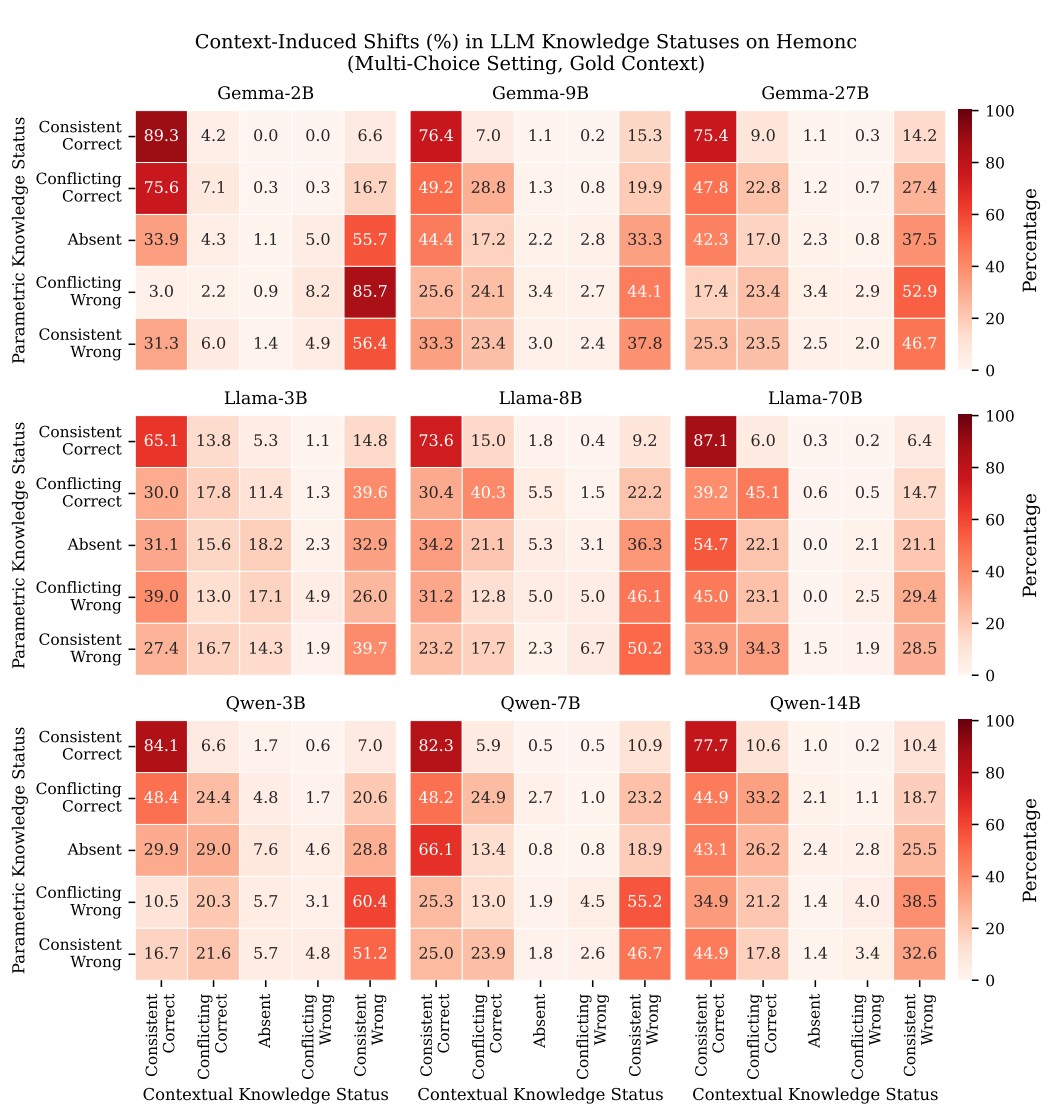

Figure 14: Shifts in knowledge status induced by gold context for each LLM on the Hemonc dataset in the multi-choice setting.

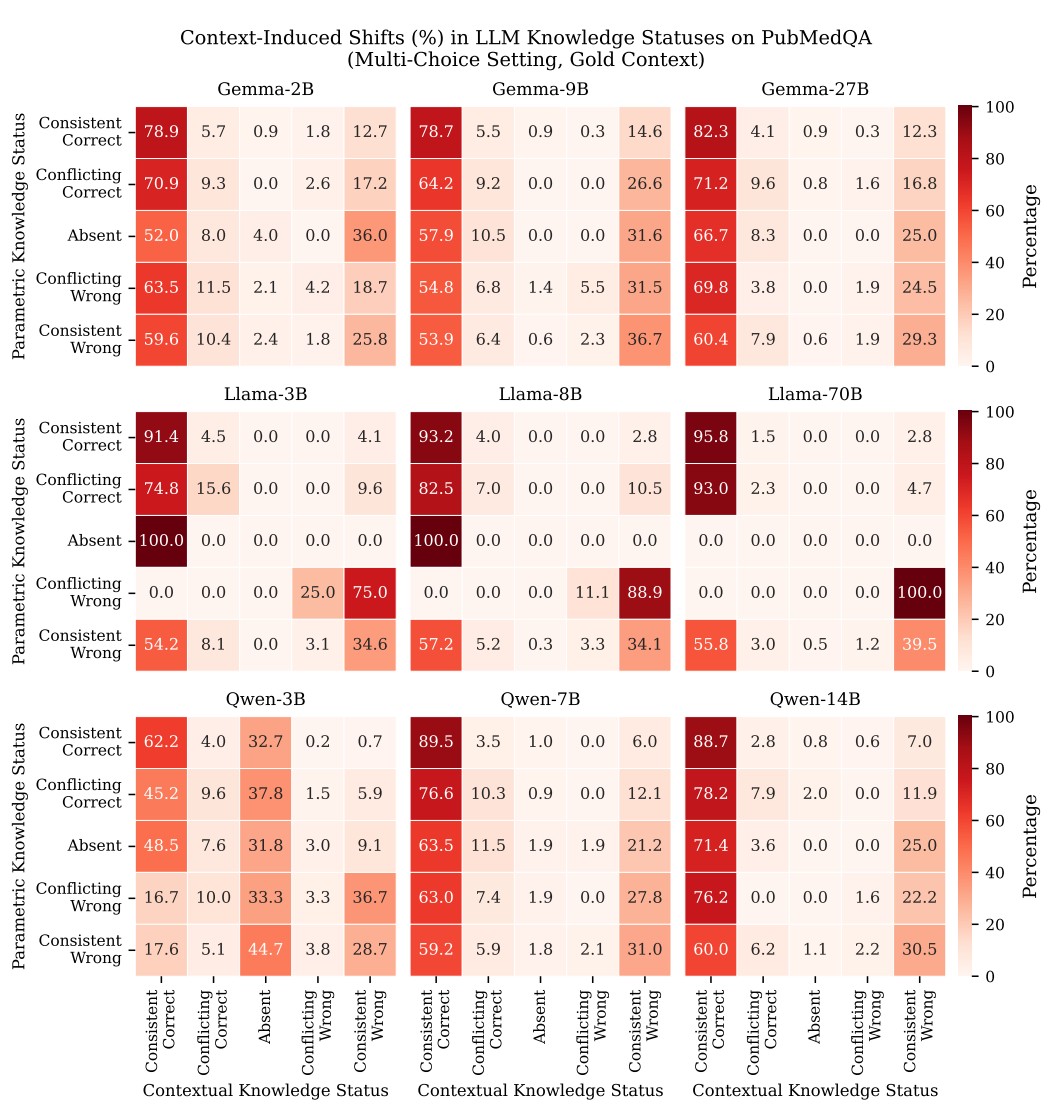

Figure 15: Shifts in knowledge status induced by gold context for each LLM on the PubMed dataset in the multi-choice setting.

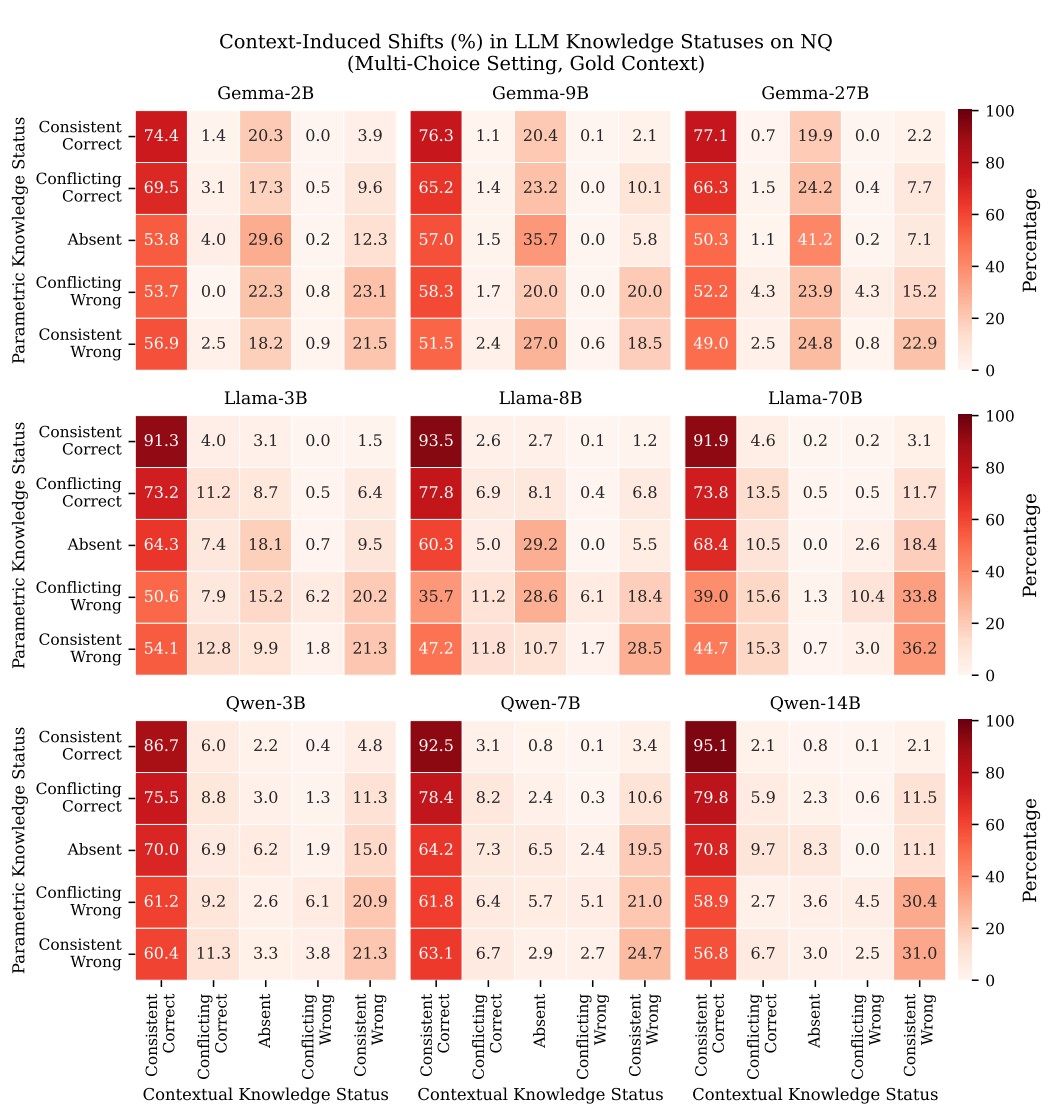

Figure 16: Shifts in knowledge status induced by gold context for each LLM on the NQ dataset in the multi-choice setting.

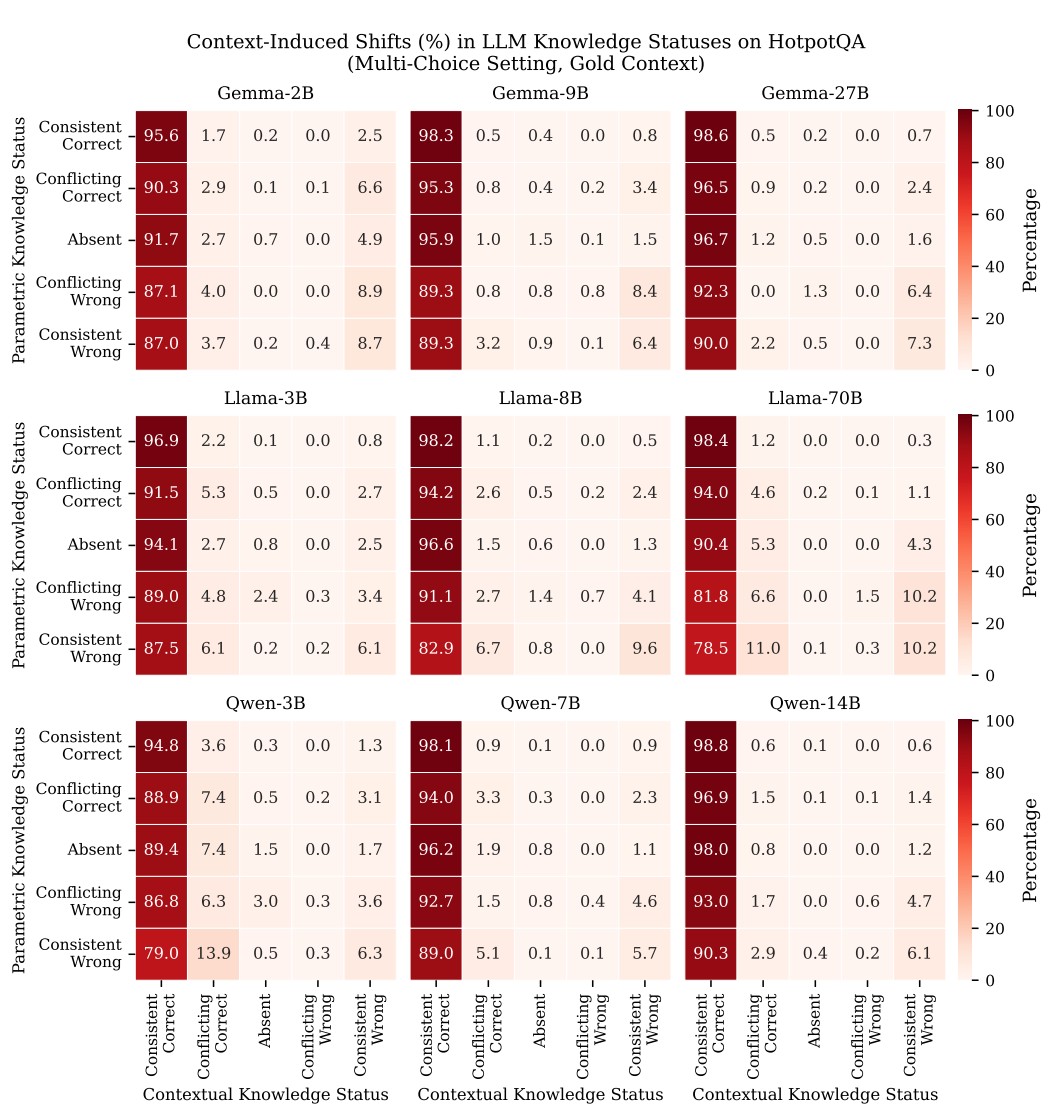

Figure 17: Shifts in knowledge status induced by gold context for each LLM on the HotpotQA dataset in the multi-choice setting.

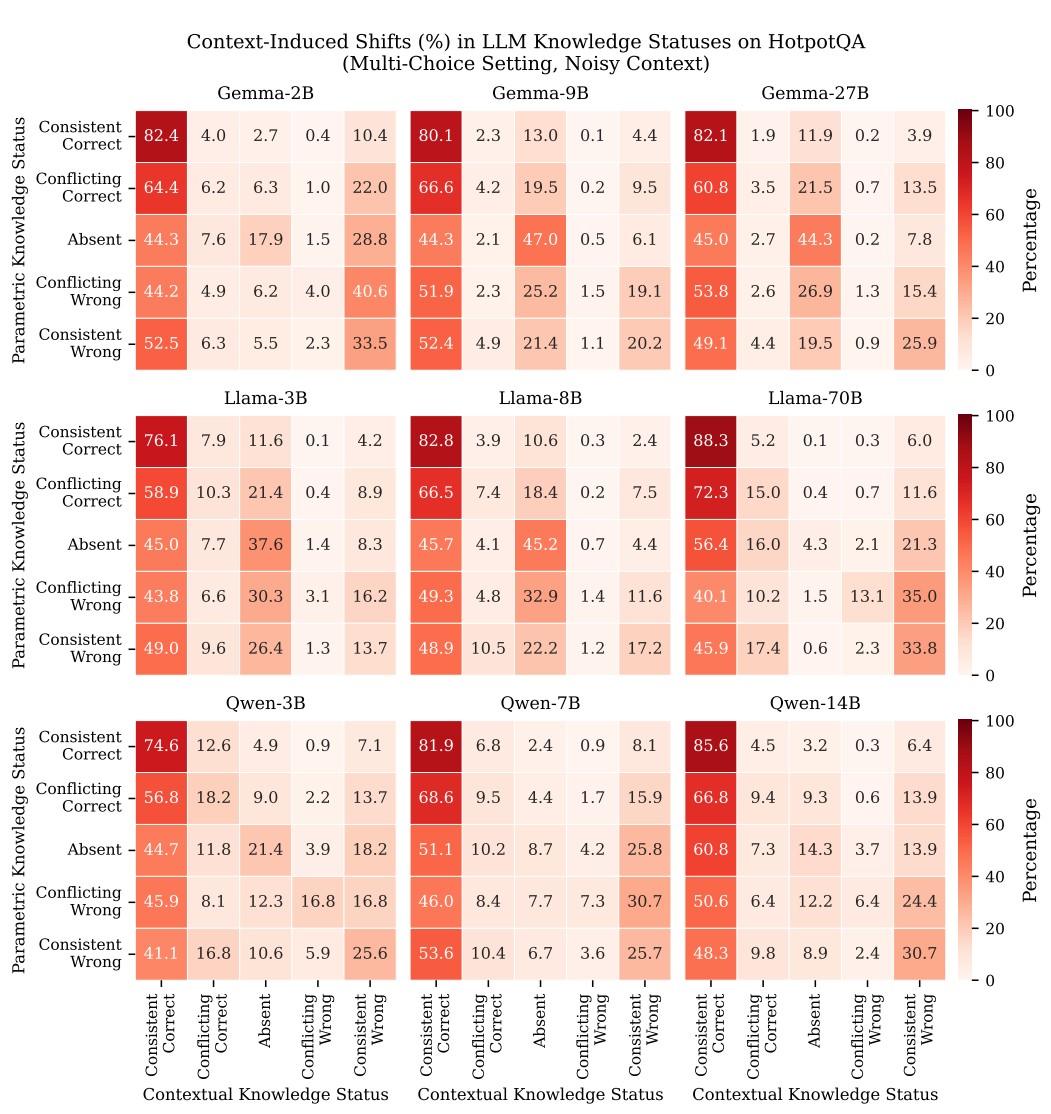

Figure 18: Shifts in knowledge status induced by noisy context for each LLM on the HotpotQA dataset in the multi-choice setting.

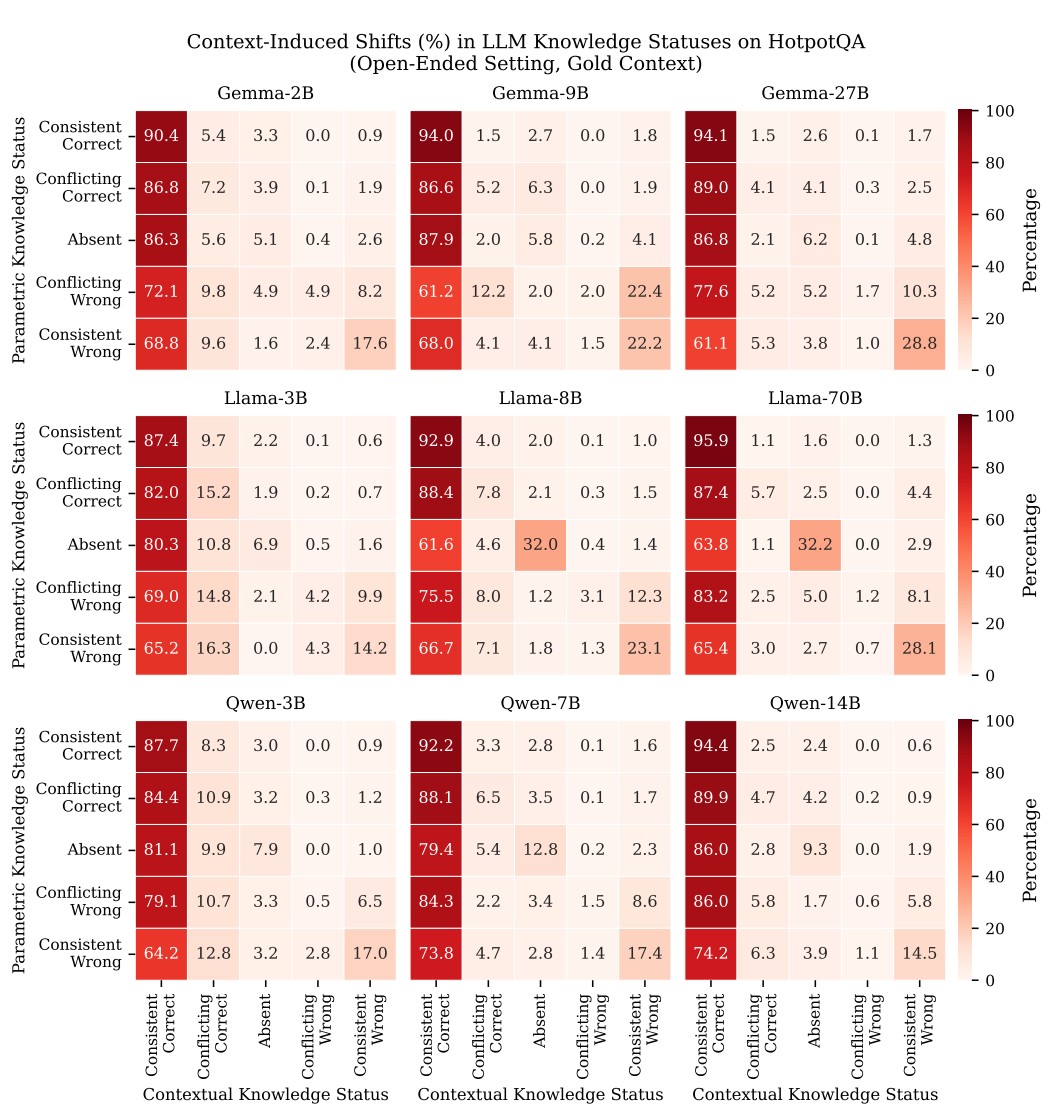

Figure 19: Shifts in knowledge status induced by gold context for each LLM on the HotpotQA dataset in the open-ended setting.

```
### Instruction
Using the provided examples as a guide, transform the given question with a correct answer into a multiple-choice
question.
Provide two additional incorrect options that are similar in type or category to the correct answer.

# Example 1
## Question: Which continent is the largest by land area?
## Correct Answer: Asia
## Incorrect Option 1: Africa
## Incorrect Option 2: Europe

# Example 2
## Question: Is the last name scott irish or scottish?
## Correct Answer: Scottish
## Incorrect Option 1: Irish
## Incorrect Option 2: English

# Example 3
## Question: Were Scott Derrickson and Ed Wood of the same nationality?
## Correct Answer: Yes
## Incorrect Option 1: No
## Incorrect Option 2: Maybe

# Your Task
## Question: {QUESTION}
## Correct Answer: {ANSWER}
```

Figure 20: Few-shot prompt used with GPT-4o to generate incorrect options.

```
### Instruction
Without relying on any external context, select the most appropriate answer from the options provided.
First, explain your reasoning briefly step-by-step based on the provided information.
Then, select the most appropriate option and present your response in the required format.

### Question:
{QUESTION}

### Choices:
Option 1: {OPTION_1}
Option 2: {OPTION_2}
Option 3: {OPTION_3}

Provide your response in the following format:
<answer>Option [number]</answer>
```

```
### Instruction
You are given some context and a multiple-choice question.
Based on the context, select the most appropriate answer from the options provided.
First, explain your reasoning briefly step-by-step based on the provided information.
Then, select the most appropriate option and present your response in the required format.

### Context:
{CONTEXT}

### Question:
{QUESTION}

### Choices:
Option 1: {OPTION_1}
Option 2: {OPTION_2}
Option 3: {OPTION_3}

Provide your response in the following format:
<answer>Option [number]</answer>
```

Figure 21: Instruction prompt for sampling model responses without (top) and with (bottom) context.

```
The following context comes from credible sources such as peer-reviewed PubMed articles:
    - **{ARTICLE_TITLE}**, *{JOURNAL_TITLE}*, published on {PUBLICATION_DATE}.

Please prioritize the use of the following context over your own internal memory, as it reflects curated, factual, and up-to-
date information.

{CONTEXT}
```

```
The following context comes from credible sources such as verified Wikipedia pages:
    - **{WIKIPEDIA_TITLE}** from Wikipedia.

Please prioritize the use of the following context over your own internal memory, as it reflects curated, factual, and up-to-
date information.

{CONTEXT}
```

Figure 22: Context augmentation prompts for inserting metadata in healthcare-related datasets (top) and general-domain datasets (bottom).

```
## Instruction
Summarize the given context.
Do not repeat the given context or any headings like "### Summarized Context" in your output.
Only return the revised summary.

## Input
### Given Context: {CONTEXT}

## Your Task
### Summarized Context:
```

```
## Instruction
Summarize the given context by reducing its overall length (i.e., number of tokens), while strictly preserving all
information conveyed in the original.
Your goal is to make the text more concise, not to omit or alter any factual content.

Follow these constraints:
1. Preserve all semantic content from the given context. Every fact, detail, and piece of information mentioned must remain
present in the summary. Nothing should be lost or distorted.
2. Maintain the naturalness and fluency of the text. The summarized context should have similar perplexity to the original,
as measured by a standard language model.
3. Ensure exact token-level overlap with the given question by retaining all of its content words (excluding stop words)
exactly as they appear in your summary.

Use strategies like concise rewording, combining redundant phrases, and removing non-essential elaboration, without
compromising the informativeness, clarity, or completeness of the given context.
Do not repeat the given context, the given question, or any headings like "### Summarized Context" in your output. Only
return the revised summary.

## Input
### Given Question: {QUESTION}
### Given Context: {CONTEXT}

## Your Task
### Summarized Context:
```

Figure 23: Instruction prompt used with GPT-4o to generate naïve (top) and constrained (bottom)
context summarization.

