# OpenReview forum: "KScope: A Framework for Characterizing the Knowledge Status of Language Models"
_NeurIPS.cc/2025/Conference — NeurIPS 2025 poster_

### Official Review · Reviewer_gP5r · 2025-06-30

**Clarity:** 2
**Significance:** 2
**Originality:** 3
**Rating:** 4
**Confidence:** 4

**Summary:**

This paper proposes KScope, a hierarchical statistical framework for classifying the knowledge status of LLMs into five categories based on correctness and consistency. It is applied to nine instruction-tuned LLMs across four datasets, enabling fine-grained analysis of knowledge behavior. The paper also explores contextual factors affecting knowledge updates and evaluates strategies like constrained summarization and credibility-based augmentation to improve LLM reliability.

**Questions:**

Please see the Weaknesses.

**Ethical Concerns:**

["NO or VERY MINOR ethics concerns only"]

**Final Justification:**

I've considered the author responses and decided to keep my original scores.

**Limitations:**

Yes

**Quality:**

3

**Strengths And Weaknesses:**

Strengths

1. Introduces a new, structured perspective on LLM knowledge states.
2. Combines statistically principled techniques (e.g., exact tests, likelihood ratio testing) with modern interpretability tools (e.g., SHAP) for credible analysis.
3. Give insights into LLMs' knowledge mechanism and underscore the importance of tailoring knowledge update strategies to different knowledge statuses in future work.


Weaknesses

1. All context passages used in the experiments are clean and supportive, which does not reflect real-world retrieval conditions where information may be noisy (e.g., irrelevant to the query)[1, 2], conflicting (e.g., contradicting the model's parametric memory), or factually incorrect. However, the authors focus exclusively on knowledge updates under supportive contexts. While this design choice is acknowledged in Section 8 (Limitations), it would be helpful to analyze how KScope performs under such imperfect or adversarial contexts.
2. Although the five knowledge states are formally defined, their separability and interpretability in borderline cases remain unclear.
    - It is unclear whether the authors manually examined ambiguous instances to verify that the taxonomy aligns with human intuition.
    - Moreover, under the multi-sample statistical testing setup, instances near decision boundaries may be prone to misclassification.

[1] https://arxiv.org/pdf/2310.01558
[2] https://arxiv.org/pdf/2404.03302

---

> ### Author Rebuttal · Authors · 2025-07-31
>
> We thank the reviewer for their constructive feedback and insightful suggestions. Please find our responses below. We will update the manuscript accordingly and would appreciate it if the reviewer could consider raising the score should these clarifications address the concerns.
>
> ---
> **1. KScope under Real-World Retrieval Conditions**
> > All context passages used in the experiments are clean and supportive, which does not reflect real-world retrieval conditions
>
> * We acknowledge that clean and supportive context may not reflect all real-world scenarios. Our initial experiments focus on such controlled settings to isolate the effect of knowledge update and to minimize variability from the retrieval process.
> * To evaluate KScope under more realistic retrieval conditions, per your suggestion, we conduct a new experiment: we apply KScope to Llama-3.1-8B-Instruct using the top ten Wikipedia paragraphs retrieved for each HotpotQA question (i.e., the fullwiki setting in HotpotQA), which may or may not include the gold supporting context.
>     * Compared to using gold passages (Figure 3), using noisy retrieval leads to a reduced success rate of updating the model’s knowledge to the consistent correct status, particularly when the LLM lacks consistent correct knowledge in its parametric memory.
>     * | Context | Consistent Correct (%) |  Conflicting Correct (%) | Absent (%) | Conflicting Wrong (%) | Consistent Wrong (%) |
> |:----|:----|:----|:----|:----|:----|
> | Supporting | 98.19 | 94.21 | 96.62 | 91.10 | 82.91 |
> | Noisy | 82.83 | 66.50 | 45.67 | 49.32 | 48.86 |
>
> ---
> **2. KScope’s Alignment with Human Intuition**
> > It is unclear whether the authors manually examined ambiguous instances to verify that the taxonomy aligns with human intuition.
>
> * To assess KScope’s alignment with human intuition, we evaluate its agreement with human judgments on the degree of knowledge conflict in LLMs.
>     * We collect 100 responses from Llama-3.1-8B-Instruct for each of 100 questions randomly sampled from HotpotQA. Based on the distribution of responses (i.e., the frequency of each option and invalid answers), three annotators label each question as “consistent”, “conflicting”, or “absent”. The inter-annotator agreement is strong (Fleiss' $\kappa = 0.61$), and we use the majority vote as the ground-truth label.
>     * To ensure a fair comparison focused on knowledge conflict (rather than correctness), we collapse KScope’s “consistent correct” and “consistent wrong” into “consistent,” and similarly collapse “conflicting correct” and “conflicting wrong” into “conflicting.”
>     * We find that KScope agrees with human judgment in 89 out of 100 cases, suggesting strong alignment with human intuition.
>
> ---
> We thank the reviewer again for their thoughtful suggestions. We hope these additional results and analyses address the concerns and further support the validity and applicability of KScope.

---

> > ### Comment · Reviewer_gP5r · 2025-08-05
> >
> > Thanks for the response. I'm leaning positive about this work, and thus intend to keep my original scores.

---

> > > ### Author Response · Authors · 2025-08-08
> > >
> > > Thank you for your response and for leaning positive on our work! We appreciate your suggestions and will update our manuscript accordingly.

---

### Official Review · Reviewer_WvzF · 2025-07-02

**Clarity:** 3
**Significance:** 3
**Originality:** 3
**Rating:** 4
**Confidence:** 3

**Summary:**

The paper introduces KScope, a hierarchical testing framework designed to characterize the knowledge status of large language models (LLMs) based on their consistency and correctness across parametric (internal) and contextual (external) knowledge modes. It proposes a taxonomy of five knowledge statuses: consistent correct, conflicting correct, absent, conflicting wrong, and consistent wrong. The framework is applied to nine LLMs across four datasets (two healthcare-related: Hemone, PubMedQA; two general: NQ, HotpotQA) to evaluate knowledge status and the impact of context on knowledge updates. Key findings include the role of context features (difficulty, relevance, familiarity) in driving successful knowledge updates and the effectiveness of constrained context summarization combined with credibility metadata.

**Questions:**

no

**Ethical Concerns:**

["NO or VERY MINOR ethics concerns only"]

**Final Justification:**

Generalization to Open-Ended Tasks address my concerns.

I would keep the positive rating.

**Limitations:**

yes

**Quality:**

3

**Strengths And Weaknesses:**

**Strength**

1. Novel Taxonomy and Framework: The introduction of a five-status taxonomy based on consistency and correctness is a significant contribution. KScope’s hierarchical testing approach (using statistical tests like binomial and multinomial tests) provides a structured method to assess LLM knowledge, addressing limitations of prior methods like entropy-based uncertainty metrics that fail to capture mode structure (e.g., distinguishing [0.45, 0.45, 0.1] from [0.6, 0.2, 0.2]).

2. Comprehensive Evaluation: The paper evaluates nine LLMs from three model families (Gemma-2, Llama-3, Qwen-2.5) across diverse datasets, ensuring robust empirical validation. The inclusion of both healthcare and general-domain datasets enhances generalizability.


**Weakness**

1. Generalization to Open-Ended Tasks: Although KScope is claimed to be applicable to open-ended questions (Section 6), the evaluation focuses on three-option classification tasks. Empirical validation on open-ended tasks would strengthen the claim of generalizability.

---

> ### Author Rebuttal · Authors · 2025-07-31
>
> We thank the reviewer for their helpful feedback. Please find our response below. We will update the manuscript accordingly and would appreciate it if the reviewer could consider raising the score if the additional results address the concern.
>
> ---
> **1. Generalization to Open-Ended Tasks**
> > Empirical validation on open-ended tasks would strengthen the claim of generalizability.
>
> * To demonstrate the generalizability of KScope to open-ended questions, we apply it to characterize the knowledge status of Llama-3.1-8B-Instruct on questions from HotpotQA, this time without providing any pre-defined answer options.
>     * Specifically, we generate $N = 100$ responses per question and semantically cluster them using a natural language inference model [1,2]. Based on the size of clusters and the number of invalid answers, we follow the procedure in Figure 2 to infer the LLM’s knowledge status.
>     * Without access to pre-defined options, the model no longer receives hints from answer options. As a result, Llama-3.1-8B-Instruct consistently exhibits absent parametric knowledge in the open-ended setting without external context, unlike in the multi-choice setting in Figure 3.
>     * However, when supporting context is provided, the distribution of knowledge statuses remains largely consistent between the open-ended and multi-choice settings.
>     * | w/ context? | w/ pre-defined options? | Consistent Correct (%) |  Conflicting Correct (%) | Absent (%) | Conflicting Wrong (%) | Consistent Wrong (%) |
> |:----|:----|:----|:----|:----|:----|:----|
> | No (Parametric) | Yes (Multi-choice) | 46.15 | 16.10 | 23.22 | 2.39 | 12.14 |
> | No (Parametric) | No (Open-Ended) | 0.00 | 0.00 | 100.00 | 0.00 | 0.00 |
> | Yes (Contextual) | Yes (Multi-choice) | 95.16 | 2.16 | 0.46 | 0.05 | 2.17 |
> | Yes (Contextual) | No (Open-Ended) | 92.68 | 0.51 | 1.98 | 0.10 | 4.74 |
>
> [1] He, Pengcheng, et al. "Deberta: Decoding-Enhanced Bert With Disentangled Attention." ICLR (2021).
> [2] Reimers, Nils, et al. "Sentence-BERT: Sentence Embeddings using Siamese BERT-Networks." EMNLP (2019).
>
> ---
> We thank the reviewer again for the constructive suggestion. We hope the added open-ended results address the concern and further support the generalizability of our approach.

---

> > ### Comment · Reviewer_WvzF · 2025-08-05
> > **Response**
> >
> > Generalization to Open-Ended Tasks address my concerns.
> >
> > Thanks for your response. I would keep the positive rating.

---

> > > ### Author Response · Authors · 2025-08-08
> > >
> > > Thank you for your constructive feedback and for maintaining a positive rating! We will incorporate your suggestions into our revised manuscript.

---

### Official Review · Reviewer_Fosa · 2025-07-02

**Clarity:** 4
**Significance:** 4
**Originality:** 3
**Rating:** 5
**Confidence:** 4

**Summary:**

This paper introduces a framework to measure when language models give consistent and reliable answers KScope. By comparing the answers from variations of questions, they apply statistical tests to understand whether a model is confident or uncertain. They evaluate this method on four question-answering datasets, using nine different open-source language models, and provide taxonomy of knowledge status.

**Questions:**

Main Questions:
- **(Q1)** Could you please add comparisons to some baselines of uncertainty metrics to show whether KScope offers a meaningful improvement over cheaper methods? How are these metrics related to KScope?
- **(Q2)** Could you comment whether paraphrasing and CoT introduces correlated outputs, which is not expected for tests assumptions?
- **(Q3)** Did you think of some real-world test for the method to produce improved downstream task performance?

Minor Questions:
- **(Q4)** I think I didn't understand how to use your method for open-ended generation. Could you please elaborate more on such application and maybe provide some experiments?

**Ethical Concerns:**

["NO or VERY MINOR ethics concerns only"]

**Final Justification:**

The authors have addressed all major concerns raised in the initial review. Mainly, they provide a comparison to semantic entropy baselines, demonstrating stronger alignment with human judgment. Their explanation of sampling independence is reasonable. The method is well-motivated and supported by strong empirical results.

I find the overall contribution strong, and vote for acceptance.

**Limitations:**

Yes

**Quality:**

3

**Strengths And Weaknesses:**

Strengths:
- **(S1)** Systematic framework: KScope is a well-structured and clearly described framework that utilise formal statistical testing and paraphrasing with CoT. This combination is novel and goes beyond traditional single-shot confidence estimation.
- **(S2)** Extensive empirical evaluation: authors evaluate KScope across 4 datasets and 9 LLMs of varying sizes and families. This broad evaluation improves generalizability. Moreover, authors carefully design their experiments and methodology
- **(S3)** Insightful analysis and ablations: authors provide useful analyses on error types, model scaling trends, and the effects of interventions like summaries or distractor quality

Weaknesses:
- **(W1)** No comparison with existing uncertainty/conflict measures: there is no comparison of KScope with prior simpler or cheaper methods like entropy-based confidence or others. This makes it hard to judge if the added complexity is really necessary.
- **(W2)** Statistical concerns in sampling: the framework assumes that samples (paraphrases and CoTs) are independent for hypothesis testing, but in practice, as I believe, they are strongly correlated (e.g., same base question, shared decoding artifacts). This may lead to overly optimistic significance results
- **(W3)** No real-world use case evaluation: the paper shows that KScope can identify unreliable answers, but it does not demonstrate this resulting in improved downstream task performance

---

> ### Author Rebuttal · Authors · 2025-07-31
>
> We thank the reviewer for their thoughtful and constructive feedback. Please find our point-by-point responses below. We will incorporate these clarifications and results into the revision, and would be grateful if the reviewer could consider raising the score if the concerns are addressed.
>
> ---
> **1. Comparison Against Existing Uncertainty Measures**
> > No comparison of KScope with prior simpler or cheaper methods like entropy-based confidence or others.
>
> * To assess the effectiveness of KScope relative to semantic entropy [1, 2], we evaluate how well each method aligns with human judgment on the degree of knowledge conflict in LLMs.
>     * We collect 100 responses from Llama-3.1-8B-Instruct for each of 200 questions randomly sampled from HotpotQA. Based on the distribution of responses (i.e. the frequency of each option and invalid answers), three annotators label each question as “consistent”, “conflicting”, or “absent”. The inter-annotator agreement is strong (Fleiss' $\kappa = 0.61$), and we use the majority vote as the ground-truth label.
>     * We split the questions into equal-sized training and test sets. For the entropy baseline, we compute semantic entropy per question using the 100 responses and train a logistic regression model using the training set. The model then predicts conflict status on the test set.
>     * In contrast, KScope requires no training or threshold tuning. To align the comparison with the task of measuring knowledge conflict (rather than correctness), we collapse KScope’s “consistent correct” and “consistent wrong” into “consistent,” and similarly collapse “conflicting correct” and “conflicting wrong” into “conflicting.”
>     * As shown below, KScope achieves stronger agreement with human judgment than semantic entropy.
>     * | Method | Accuracy |
> |:----|:----|
> | KScope | 0.890 |
> | Semantic Entropy | 0.720 |
>
> [1] Kuhn, Lorenz, et al. "Semantic Uncertainty: Linguistic Invariances for Uncertainty Estimation in Natural Language Generation." ICLR (2023).
> [2] Marjanovic, Sara, et al. "DYNAMICQA: Tracing Internal Knowledge Conflicts in Language Models." EMNLP Findings (2024).
>
> ---
> **2. Real-World Use Cases**
> > Did you think of some real-world test for the method to produce improved downstream task performance?
>
> * Compared to entropy-based metrics [1, 2], KScope offers an interpretable characterization of LLM knowledge status without the need for threshold calibration. By producing a distribution over knowledge statuses (e.g., as shown in Figure 3), practitioners can better assess model reliability and adapt usage accordingly in high-stakes or domain-specific settings.
> * Additionally, Sections 6 and 7 show that KScope’s outputs can guide context augmentation, significantly improving the success rate of knowledge updates across different LLMs.
>
> ---
> **3. Generalization to Open-Ended Tasks**
> > Elaborate more on open-ended generation and maybe provide some experiments.
>
> * To demonstrate KScope’s generalizability to open-ended questions, we apply it to HotpotQA without supplying predefined answer options.
>     * We generate $N = 100$ responses per question using Llama-3.1-8B-Instruct, cluster them semantically using a natural language inference model [3, 4], and follow the procedure in Figure 2 to determine the LLM’s knowledge status.
>     * Without predefined options, the model lacks informational hints, and thus consistently exhibits absent parametric knowledge in the open-ended setting without external context, unlike in the multi-choice setting in Figure 3.
>     * However, once supporting context is provided, the resulting knowledge status distribution closely matches that of the multi-choice setting.
>     * | w/ context? | w/ pre-defined options? | Consistent Correct (%) |  Conflicting Correct (%) | Absent (%) | Conflicting Wrong (%) | Consistent Wrong (%) |
> |:----|:----|:----|:----|:----|:----|:----|
> | No (Parametric) | Yes (Multi-choice) | 46.15 | 16.10 | 23.22 | 2.39 | 12.14 |
> | No (Parametric) | No (Open-Ended) | 0.00 | 0.00 | 100.00 | 0.00 | 0.00 |
> | Yes (Contextual) | Yes (Multi-choice) | 95.16 | 2.16 | 0.46 | 0.05 | 2.17 |
> | Yes (Contextual) | No (Open-Ended) | 92.68 | 0.51 | 1.98 | 0.10 | 4.74 |
>
> [3] He, Pengcheng, et al. "Deberta: Decoding-Enhanced Bert With Disentangled Attention." ICLR (2021).
> [4] Reimers, Nils, et al. "Sentence-BERT: Sentence Embeddings using Siamese BERT-Networks." EMNLP (2019).
>
> ---
> **4. Statistical Independence in Sampling**
> > Could you comment whether paraphrasing and CoT introduces correlated outputs?
>
> * We thank the reviewer for raising this point. Since we perform paraphrasing independently for each variation, and similarly generate CoT inferences independently, we assume that the resulting samples are sufficiently diverse and statistically independent for the purpose of characterizing knowledge status.
>
> ---
> We thank the reviewer again for their thoughtful feedback. We hope these additional results and clarifications address the raised concerns and further support the significance and applicability of KScope.

---

> > ### Comment · Reviewer_Fosa · 2025-08-03
> >
> > Thank you for the detailed and thoughtful rebuttal! The new experiments directly address my concerns. I appreciate the clarifications and the additional empirical evidence. I am raising my score.

---

> > > ### Author Response · Authors · 2025-08-08
> > >
> > > Thank you for your thoughtful engagement and for raising your score! We are glad that the new experiments addressed your concerns, and we will reflect your suggestions in our revision.

---

### Official Review · Reviewer_fYMy · 2025-07-03

**Clarity:** 3
**Significance:** 3
**Originality:** 3
**Rating:** 4
**Confidence:** 3

**Summary:**

The paper studies how to deeply characterize an LLM's knowledge state regarding a specific problem. The paper proposes a novel taxonomy of knowledge status, which categorizes the knowledge state of LLMs into five distinct classifications based on the dimensions of "Consistency" and "Correctness". The paper then introduces KScope,  a framework that systematically and reproducibly classifies the
model's knowledge state into one of the aforementioned categories through a series of statistical tests, including binomial tests, multinomial tests, and likelihood ratio tests.

**Questions:**

1. The test result of a model's knowledge status is based on the model's behavior within the support set. Since the support set is
human-made or model-generated, will the result be influenced by the support set? For example, have you analyzed the
impact of "difficult" negative samples and "simple" negative samples on the classification results of knowledge states?
Besides, if the question's support set is (A.dog B.cat) and the model's internal knowledge is conflicting which is (cat,
duck), since the model is forced to choose one among the support set, it will prefer 'B.cat' and will be classified as
'consistent correct'. However, if we replace the 'A.dog' with 'A.duck', then the model will prefer both of them, and will be
classified as 'conflicting correct'.

2. The paper defines "knowledge absent" as the model exhibiting a uniform distribution on the support set. While this is
indeed an expression of "not knowing," it may not be the only or most common manifestation. When lacking knowledge,
LLMs may also persistently output rejection responses like "I don't know" or "unable to answer," or generate
hallucinations beyond the support set. Although the first step of KScope tests the significance of invalid answers, it seems
to categorize these complex "absent" behaviors under a general "knowledge deficiency" state. Is it possible for the
framework to further distinguish these different "absent" states, given that they may require distinct intervention
strategies?

**Ethical Concerns:**

["NO or VERY MINOR ethics concerns only"]

**Final Justification:**

The rebuttal answered my question about the support set soundly and provided explanations for the relationship between refusal/hallucination and absent knowledge.

**Limitations:**

yes

**Quality:**

3

**Strengths And Weaknesses:**

Strengths:
1. The proposed five classification methods of knowledge states go beyond the previous binary divisions of
"correct/incorrect" or "conflicting/non-conflicting", providing a more detailed and insightful analytical perspective. I
believe this classification method is an important contribution to the community.

2. The design of the KScope framework is outstanding. It's based on a series of mature statistical tests, which makes its
conclusions highly scientific and reliable. This approach of applying statistical theory to LLM behavior analysis provides
a reproducible and standardized operational process for subsequent research.

Weaknesses:
1. This method requires sampling N = 100 outputs for each question to construct a stable empirical distribution. This is
computationally expensive for large models and large-scale evaluations. Although this is necessary for the validity of
statistical tests, it may limit the practical application of KScope in scenarios that require the analysis of massive data.

2. For the NQ and HotpotQA datasets, the authors generated two additional incorrect options using GPT-4o and transformed
them into a three-choice task. The quality of these AI-generated "distractions" may significantly affect the judgment of the
model's knowledge state. A very obvious incorrect option and a wrong option that is very similar to the correct answer
pose completely different challenges to the model.

---

> ### Author Rebuttal · Authors · 2025-07-31
>
> We thank the reviewer for their thoughtful comments and valuable suggestions. Please find our responses below. We will update the manuscript accordingly and would be grateful if the reviewer could consider raising the score if these clarifications address the concerns.
>
> ---
> **1. Influence of Support Sets**
> > Since the support set is human-made or model-generated, will the result be influenced by the support set?
> > The quality of these AI-generated "distractions" may significantly affect the judgment of the model's knowledge state.
>
> * We acknowledge that our empirical characterization of LLMs’ knowledge status in KScope is based on both the question and its associated pre-defined support set, since exhaustively enumerating all possible support sets is computationally infeasible.
> * To examine the influence of support sets, we further apply KScope to open-ended questions without providing any pre-defined options.
>     * Specifically, we generate $N = 100$ responses for each HotpotQA question using Llama-3.1-8B-Instruct, and cluster them semantically using a natural language inference model [1,2]. Based on the size of clusters and the number of invalid answers, we follow the procedure in Figure 2 to infer the model’s knowledge status.
>     * In the absence of pre-defined support sets, the model no longer receives hints from answer options. As a result, Llama-3.1-8B-Instruct consistently exhibits absent parametric knowledge in the open-ended setting without external context, unlike in the multi-choice setting in Figure 3.
>     * However, when supporting context is provided, the distribution of knowledge statuses remains largely consistent between the open-ended and multi-choice settings.
>     * | w/ context? | w/ pre-defined options? | Consistent Correct (%) |  Conflicting Correct (%) | Absent (%) | Conflicting Wrong (%) | Consistent Wrong (%) |
> |:----|:----|:----|:----|:----|:----|:----|
> | No (Parametric) | Yes (Multi-choice) | 46.15 | 16.10 | 23.22 | 2.39 | 12.14 |
> | No (Parametric) | No (Open-Ended) | 0.00 | 0.00 | 100.00 | 0.00 | 0.00 |
> | Yes (Contextual) | Yes (Multi-choice) | 95.16 | 2.16 | 0.46 | 0.05 | 2.17 |
> | Yes (Contextual) | No (Open-Ended) | 92.68 | 0.51 | 1.98 | 0.10 | 4.74 |
>
> [1] He, Pengcheng, et al. "Deberta: Decoding-Enhanced Bert With Disentangled Attention." ICLR (2021).
> [2] Reimers, Nils, et al. "Sentence-BERT: Sentence Embeddings using Siamese BERT-Networks." EMNLP (2019).
>
> ---
> **2. Relationship between Refusal/Hallucination and Absent Knowledge**
> > The paper defines "knowledge absent" as the model exhibiting a uniform distribution on the support set.
> > Although the first step of KScope tests the significance of invalid answers, it seems to categorize these complex "absent" behaviors under a general "knowledge deficiency" state.
>
> * We argue that when a model lacks sufficient knowledge to a question, it may (1) refuse to respond, (2) hallucinate invalid answers, or (3) generate valid answers at random. Hence, we group all three behaviors under the status of absent knowledge in KScope.
> * To further distinguish between refusal and hallucination, we need to split Step 1 of KScope, which tests for the significance of invalid answers, and conduct separate statistical tests for these two types of behaviors. We leave this extension to future work.
>
> ---
> We thank the reviewer again for their insightful feedback. If our clarifications address the concerns, we would be grateful if the reviewer could consider raising the score.

---

> > ### Comment · Reviewer_fYMy · 2025-08-02
> >
> > Thanks for your response. I would keep the rating and vote for acceptance of the paper.

---

> > > ### Author Response · Authors · 2025-08-08
> > >
> > > Thank you for supporting our submission! We appreciate your feedback and will incorporate your suggestions into our revision.

---

### Note · Authors · 2025-08-13

We sincerely thank all reviewers for their constructive feedback, and the ACs, SACs, and PCs for their thoughtful consideration of our submission.

**Recap of our contribution:**

Our work proposes KScope, a hierarchical statistical framework with a novel five-status taxonomy for characterizing LLM knowledge status. We evaluate nine LLMs across four datasets, identify key context features that drive successful updates, and demonstrate that constrained summarization with credibility metadata substantially improves update success rates and generalizes across models.

We are grateful that reviewers recognized the novelty of our taxonomy (**fYMy, WvzF**), the principled and reproducible statistical design of KScope (**fYMy, gP5r**), the breadth and rigor of our evaluation (**Fosa, WvzF**), the insightful analyses and ablations (**Fosa**), and the practical downstream relevance of our findings (**Fosa, gP5r**).

---
**Addressing concerns:**

We have made every effort to address all concerns:
1. **Generalization beyond predefined support sets (fYMy, Fosa, WvzF)**: We added open-ended experiments showing consistent trends with and without options.
2. **Comparison with uncertainty baselines (Fosa)**: We provided new results showing KScope surpasses semantic entropy in agreement with human judgment.
3. **Performance under noisy or adversarial context (gP5r)**: We evaluated KScope with noisy retrieval and reported the impact on update success.
4. **Alignment with human intuition (fYMy, gP5r)**: We quantified agreement between KScope and human annotations (89%).

In light of these additions and clarifications, we are confident that all raised points have been properly addressed.

---
We believe our contributions offer substantial value to the community, providing a structured and interpretable way to assess LLM knowledge and for effective knowledge updates. We respectfully ask that these points be considered in the final decision, and we will incorporate all suggestions into the final version.

Thank you again for your time and consideration.

Warm regards,
Authors

---

### Decision · Program_Chairs · 2025-09-17

**Decision:**

Accept (poster)

**Comment:**

This paper tackles the important and challenging problem of understanding what LLMs truly "know." The proposed KScope framework is methodologically sound, leveraging established statistical tests to provide a more granular and principled characterization of LLM knowledge than prior work. The reviewers rightly identified the novelty of the five-status taxonomy and the rigor of the experimental design as major strengths.

The reviewers raised several concerns about the paper, such as generalizability beyond multiple-choice questions (Reviewers fYMy, Fosa, WvzF), comparison to baselines such as semantic entropy (Reviewer Fosa), realism of context beyond clean, supportive context (Reviewer gP5r) and alignment with human intuition (Reviewer gP5r).

However, the authors' rebuttal was exemplary. They conducted a significant number of new experiments that directly and thoroughly addressed every major concern raised by the reviewers. The authors have demonstrated an impressive commitment to improving their work during the review process. The resulting paper is substantially stronger and represents a solid contribution to the field of LLM evaluation. While the computational cost of the method (requiring multiple samples) is a limitation, the insights it provides are valuable. The paper is well-executed, the claims are now better supported by the new experiments, and it will be of interest to the NeurIPS community.

I am therefore confident in recommending this paper for acceptance as a poster. The authors should be strongly encouraged to integrate their rebuttal experiments and analyses into the final camera-ready version.